# Interpreting Global Perturbation Robustness of Image Models using Axiomatic Spectral Importance Decomposition

**Róisín Luo (Jiaolin Luo)**        *j.luo2@universityofgalway.ie*

**James McDermott**        *james.mcdermott@universityofgalway.ie*

**Colm O'Riordan**        *colm.oriordan@universityofgalway.ie*

*SFI Centre for Research Training in Artificial Intelligence*
*School of Computer Science, University of Galway*
*Galway, H91 TK33, Ireland*

**Reviewed on OpenReview:** *https://openreview.net/forum?id=uQYomAuo7M*

## Abstract

Perturbation robustness evaluates the vulnerabilities of models, arising from a variety of perturbations, such as data corruptions and adversarial attacks. Understanding the mechanisms of perturbation robustness is critical for global interpretability. We present a model-agnostic, global mechanistic interpretability method to interpret the perturbation robustness of image models. This research is motivated by two key aspects. First, previous global interpretability works, in tandem with robustness benchmarks, *e.g.* mean corruption error (mCE), are not designed to directly interpret the mechanisms of perturbation robustness within image models. Second, we notice that the spectral signal-to-noise ratios (SNR) of perturbed natural images exponentially decay over the frequency. This power-law-like decay implies that: Low-frequency signals are generally more robust than high-frequency signals – yet high classification accuracy can not be achieved by low-frequency signals alone. By applying Shapley value theory, our method axiomatically quantifies the predictive powers of robust features and non-robust features within an information theory framework. Our method, dubbed as **I-ASIDE** (**I**mage **A**xiomatic **S**pectral **I**mportance **D**ecomposition **E**xplanation), provides a unique insight into model robustness mechanisms. We conduct extensive experiments over a variety of vision models pre-trained on ImageNet, including both convolutional neural networks (*e.g. AlexNet*, *VGG*, *GoogLeNet/Inception-v1*, *Inception-v3*, *ResNet*, *SqueezeNet*, *RegNet*, *MnasNet*, *MobileNet*, *EfficientNet*, etc.) and vision transformers (*e.g. ViT*, *Swin Transformer*, and *MaxViT*), to show that **I-ASIDE** can not only **measure** the perturbation robustness but also **provide interpretations** of its mechanisms.

## 1 Introduction

Image modeling with deep neural networks has achieved great success (Li et al., 2021; Khan et al., 2022; Han et al., 2022). Yet, deep neural networks are known to be vulnerable to perturbations. For example, the perturbations may arise from corruptions and adversarial attacks (Goodfellow et al., 2014; Hendrycks & Dietterich, 2019; Szegedy et al., 2013), etc. Perturbation robustness, henceforth referred to as robustness, characterizes a crucial intrinsic property of models (Hendrycks & Dietterich, 2019; Bai et al., 2021; Goodfellow et al., 2014; Silva & Najafirad, 2020).

Robustness mechanisms refer to the mechanisms which lead to robustness in models. The study of robustness mechanisms aims to answer the question '**why some models are more robust than others**' (Lipton, 2018; Zhang et al., 2021; Bereska & Gavves, 2024). The causes within this question can arise from multifarious

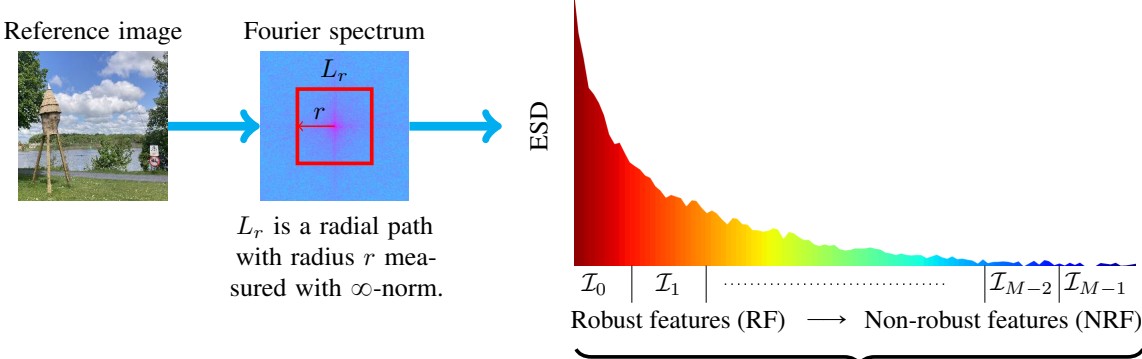

The 2D spectrum is mapped into 1D radial spectrum and further divided into $M$ bands which are denoted from $\mathcal{I}_0$ to $\mathcal{I}_{M-1}$.

**Figure 1:** Power-law-like energy spectral density (ESD) distribution of natural images over the frequency. The signal spectrum is divided into $M$ bands (from $\mathcal{I}_0$ to $\mathcal{I}_{M-1}$). Each spectral band is a robustness band.

aspects, including architecture designs (Zhou et al., 2022), data annotations, training methods (Pang et al., 2020), inferences, etc. For example, noisy labels can arise from data annotations (Frénay & Verleysen, 2013; Wei et al., 2021); adversarial attacks often take place in inferences. This research provides a unified view regarding the mechanisms of image model robustness on spectra. We only focus on global interpretation of robustness mechanisms of image models. Our method is primarily motivated by two key aspects, as detailed below.

Despite the substantial advances in previous global interpretability works (Covert et al., 2020; Kolek et al., 2022), in tandem with the wide adoption of empirical robustness benchmarks (Hendrycks & Dietterich, 2019; Zheng et al., 2016; Zhang et al., 2021), these methods are not designed to provide global interpretations regarding model robustness. For example, SAGE (Covert et al., 2020), a global interpretability work, can attribute the decisive pixel features in decision-makings. Yet, the importance of decisive pixel features fails to interpret the global robustness. Although the robustness of image models can be quantified by mean corruption errors (mCE) (Hendrycks & Dietterich, 2019) or the distances in feature spaces between clean and perturbed image pairs (Zheng et al., 2016; Zhang et al., 2021), these scalar metrics often fall short in interpreting the underlying 'why' question. This gap prompts us to provide global mechanistic interpretations of perturbation robustness.

Motivation also arises from the robustness characterization of the spectral signals within natural images. Images can be represented as spectral signals (Körner, 2022; Sherman & Butler, 2007). The signal-to-noise ratios (SNR) (Sherman & Butler, 2007) of spectral signals can be used to characterize the signal robustness with respect to perturbations. Our later empirical study, as illustrated in Figure 2, suggests that the spectral SNRs of perturbed natural images decay over the frequency with a power-law-like distribution. We refer to the *spectral SNR* as the ratio of the point-wise energy spectral density (ESD) (Stoica et al., 2005) of spectral signal to noise (*i.e.* perturbation):

$$SNR(r) := \frac{ESD_r(x)}{ESD_r(\Delta x)} \tag{1}$$

where $r$ is a radial frequency point, $x$ is an image, $\Delta x$ is the perturbation, $ESD_r(\cdot)$ gives the point-wise energy density at $r$. The $ESD_r(\cdot)$ is defined as:

$$ESD_r(x) := \frac{1}{|L_r|} \cdot \sum_{(u,v)\in L_r} |\mathscr{F}(x)(u,v)|^2 \tag{2}$$

where $L_r$ denotes a circle as illustrated in Figure 1, $|L_r|$ denotes the circumference of $L_r$, and $\mathscr{F}(x)$ denotes the 2D Discrete Fourier Transform (DFT) of $x$. Readers can further refer to more details in Appendix B.1.

Why do the spectral SNRs of some corruptions and adversarial attacks exhibit a power-law-like decay over the frequency? We surmise that the ESDs of many perturbations are often not power-law-like, while the ESDs of natural images are power-law-like empirically, as shown in Figure 1. For example, the spatial perturbation drawn from $\mathcal{N}(0, \sigma^2)$ (*i.e.* white noise) has a constant ESD: $ESD_r(\Delta x) = \sigma^2$. In Figure 2, we characterize the spectral SNR distributions of perturbed images. We set the energy of perturbations to 10% of the energy of the clean image for a fair comparison. The perturbation sources include corruptions (Hendrycks & Dietterich, 2019) and adversarial attacks (Szegedy et al., 2013; Tsipras et al., 2018). We notice that the SNR distributions are also power-law-like over the frequency. We refer to *spectral signals* as *spectral features* or simply as *features* if without ambiguity. This power-law-like SNR decay suggests an empirical feature robustness prior: **Low-frequency features are robust features (RFs) while high-frequency features are non-robust features (NRFs)**. The experiments in Figure 3 demonstrate that models trained with low-frequency signals exhibit higher robustness compared to the models trained with high-frequency signals. This also echoes with our earlier claim "low-frequency signals are generally more robust than high-frequency signals – yet high classification accuracy can not be achieved by low-frequency signals alone".

**Contributions**. By applying the Shapley value theory framework (Shapley, 1997; Roth, 1988; Hart, 1989), we are able to assess the predictive powers (Zheng & Agresti, 2000) of RFs and NRFs. Further leveraging a specifically designed characteristic function of the Shapley value theory framework, the predictive powers are assessed on the basis of their information gains. **I-ASIDE** uses information theory within the Shapley value theory framework, for interpreting robustness mechanisms, as detailed in Section 3. We claim our major contributions as:

- We propose a model-agnostic, global interpretability method, for interpreting robustness mechanisms of image models, through the lens of the predictive powers of robust features and non-robust features;

- We analyze the robustness mechanisms of image models within information theory on spectra;

- We showcase a case study that **I-ASIDE** has the potential to interpret how supervision noise levels affect model robustness.

## 2 Notations

**Image classifier**. The primary task of an image classifier is to predict the probability distributions over discrete classes for given images. We use $Q(y|x; \theta) : (x, y) \mapsto [0, 1]$ to denote a classifier in the form of conditional probability. The $Q$ predicts the probability that an image $x$ is of class $y$. The $\theta$ are the parameters. For brevity, we ignore the parameter $\theta$. For example, we denote $Q(y|x; \theta)$ as $Q(y|x)$.

**Dataset and annotation**. We use a tuple $\langle \mathcal{X}, \mathcal{Y} \rangle$ to denote an image classification dataset, where $\mathcal{X}$ is the image set and $\mathcal{Y}$ is the label set. We use $|\mathcal{Y}|$ to denote the number of classes (*i.e.* the cardinality of set $\mathcal{Y}$). The annotation task of image classification datasets is to assign each image with a discrete class probability distribution. We use $P(y|x)$ to denote the ground-truth probability that an image $x$ is assigned as a class $y$. We use $P(x)$ to denote the probability of $x$ in set $\mathcal{X}$. We use $P(y)$ to denote the probability of $y$ in set $\mathcal{Y}$. In class-balanced datasets, $P(y) = \frac{1}{|\mathcal{Y}|}$.

## 3 Method

**High-level overview**. We apply Shapley value theory for axiomatically assigning credits to spectral bands. Within this framework, the specially devised characteristic function measures the information gains of spectral bands. **I-ASIDE** interprets robustness mechanisms using this axiomatic framework with the information theory.

**Problem formulation**. Quantifying the predictive powers of features can be viewed as a *value* decomposition problem. In this research, the *value* is the information quantities that the features contribute to decisions.

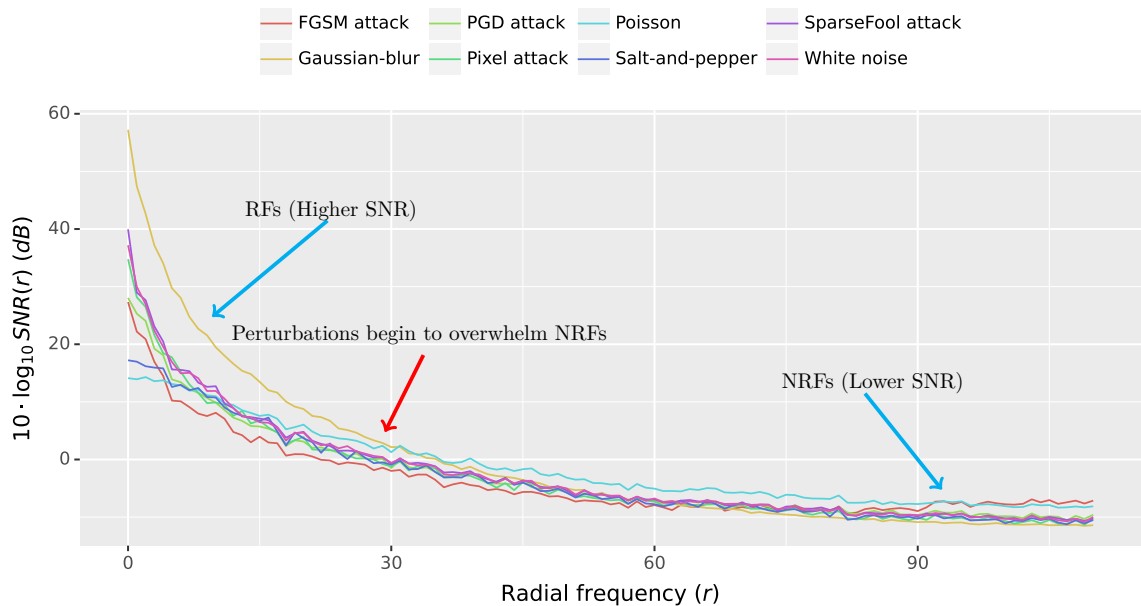

**Figure 2:** Spectral SNR characterization with multiple corruptions and adversarial attacks. The corruptions include: white noise, Poisson noise, Salt-and-pepper noise, and Gaussian blur. The adversarial attacks include: FGSM (Goodfellow et al., 2014), PGD (Madry et al., 2017), SparseFool (Modas et al., 2019) and Pixel (Pomponi et al., 2022). We set the perturbation energy to 10% of the energy of the reference image. The results are shown in decibels (dB) for better visualization. The dB below zero indicates that the perturbations overwhelm the spectral features.

Specifically, the *value* is in the form of the log-likelihood expectation of predictions (*i.e.* the negative cross-entropy loss). The *value* decomposition aims to assign each robustness band a predictive power such that: (1) The sums of the predictive powers are equal to the *value*, and (2) the assigned predictive powers should reflect their importance in the decisions. In the coalitional game theory, this decomposition scheme is known as an *axiomatic fairness division problem* (Roth, 1988; Hart, 1989; Winter, 2002; Klamler, 2010; Han & Poor, 2009). The fairness division must satisfy four axioms: *efficiency*, *symmetry*, *linearity* and *dummy player* (Roth, 1988). We refer to the *axiomatic fairness division* as *axiomatic decomposition*. Of the scheme, the axioms guarantee *uniqueness* and *fairness* (Aumann & Maschler, 1985; Yaari & Bar-Hillel, 1984; Aumann & Dombb, 2015; Hart, 1989; Roth, 1988). The property *fairness* refers to the principle '*equal treatment of equals*' (Yokote et al., 2019; Navarro, 2019). Shapley value theory is the unique solution satisfying the above axioms. However, the theory merely provides an abstract framework. To employ, we have to instantiate two abstracts: (1) players and coalitions, and (2) characteristic function.

**Abstract (1): players and coalitions**. The spectral bands are dubbed as *spectral players*. A subset of the spectral player set is dubbed as a *spectral coalition*. The details are as shown in Section 3.1. The $M$ spectral players can forge $2^M$ spectral coalitions. We represent the presences and absences of the spectral players as the pass-bands and stop-bands in a multi-band-pass digital signal filter (Oppenheim, 1978; Roberts & Mullis, 1987; Pei & Tseng, 1998), as shown in Section 3.2.

**Abstract (2): characteristic function**. The characteristic function is designed to measure the contributions that the coalitions contribute in decisions. The contributions of the $2^M$ coalitions are then combined to compute their marginal contributions in decisions. We specially design the characteristic function, as shown in Section 3.3 and Appendix B.3, to measure the information gains. Figure 4 shows the framework of applying the Shapley value theory. Figure 5 shows the block diagram of the spectral coalition filtering. Figure 6 shows an example of $2^M$ spectral coalitions.

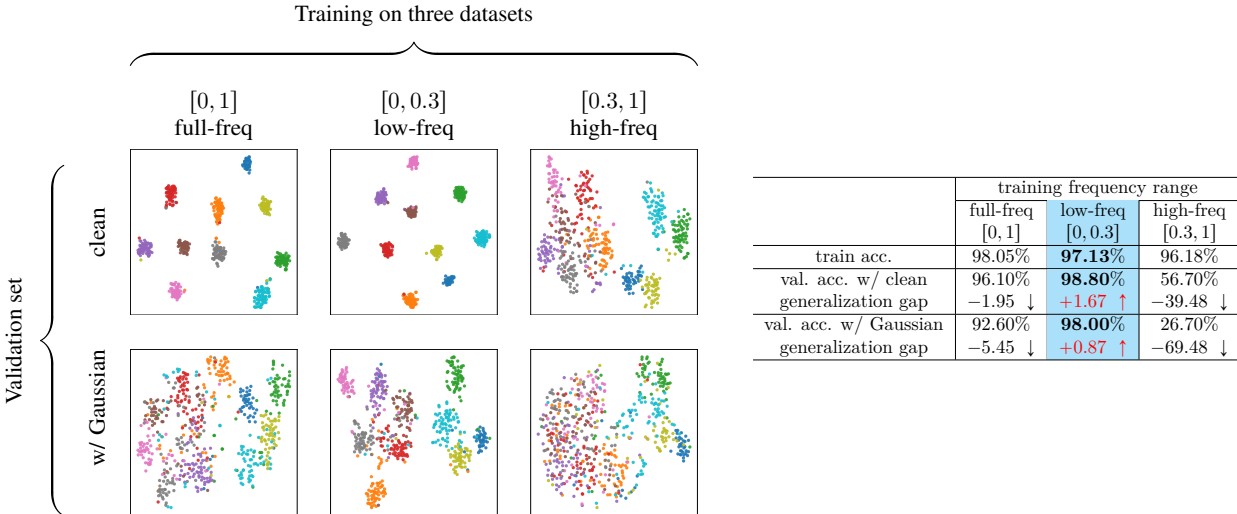

**Figure 3:** Understanding the role of spectral signals. We train a *resnet18* on three datasets derived from *STL10* (Coates et al., 2011): $[0, 1]$ contains full-frequency signals; $[0, 0.3]$ only contains low-frequency signals with a cut-off frequency by 0.3; and $[0.3, 1]$ only contains the high-frequency signal with a cut-off frequency by 0.3. In the derived datasets, the filtered high or low frequency signals are randomly replaced by the high or low frequency signals sampled from the full-frequency train set. We visualize the embeddings in the left figure with respect to clean samples and perturbed samples with Gaussian noise ($\sigma = 0.1$). The samples are from the validation set. The accuracy on both the train and validation sets is provided in the right table. We also provide generalization gaps, measured by the differences between validation accuracy and train accuracy (*i.e.* $\mathrm{acc}_{val} - \mathrm{acc}_{train}$). We have noted in that: (1) both high-frequency and low-frequency signals contain sufficient discriminative information to achieve high training accuracy; (2) high-frequency signals alone are not robust signals because they fail to generalize well from train to validation; (3) overall, low-frequency signals are more robust signals because models trained alone with them generalize better and exhibit higher robustness. We summarize these empirical observations into Assumption 3.8.

We organize the implementation details of instantiating the aforementioned two abstracts from three aspects: (1) formulating a spectral coalitional game, (2) the implementation of spectral coalitions and (3) the design of the characteristic function.

### 3.1 Spectral coalitional game

**Spectral player**. We use $\mathcal{I}_i$ (where $i \in [M] := \{0, 1, \cdots, M - 1\}$) to denote the $i$-th spectral player. The $\mathcal{I}_0$ contains the most robust features and the $\mathcal{I}_{M-1}$ contains the most non-robust features. The $M$ spectral players constitute a player set $\mathcal{I} := \{\mathcal{I}_i\}_{i=0}^{M-1}$. Figure 15 in Appendix C.1 shows two partition schemes to partition spectral bands ($\ell_\infty$ and $\ell_2$). We empirically choose $\ell_\infty$.

**Spectral coalition**. A subset $\widetilde{\mathcal{I}} \subseteq \mathcal{I}$ is referred to as the *spectral coalition*. The player set $\mathcal{I}$ is often referred to as the *grand coalition.*

**Characteristic function**. A characteristic function $v(\widetilde{\mathcal{I}}) : \widetilde{\mathcal{I}} \mapsto \mathbb{R}$ measures the contribution for a given coalition and satisfies $v(\varnothing) = 0$. In this research, the contribution of $\widetilde{\mathcal{I}}$ is measured in the form of the log-likelihood expectation of the predictions by the $Q$, in which the input images only contain the signals present in the $\widetilde{\mathcal{I}}$. We show that this design of $v$ theoretically measures how much information the $Q$ uses from the features in the $\widetilde{\mathcal{I}}$ for decisions.

**Shapley value**. A spectral coalitional game $(\mathcal{I}, v)$ is defined on a spectral player set $\mathcal{I}$ equipped with a characteristic function $v$. The weighted marginal contribution of a spectral player $\mathcal{I}_i$ over all possible coalitions is referred to as the Shapley value of the spectral player $\mathcal{I}_i$. We use $\psi_i(\mathcal{I}, v)$ to represent the Shapley value of

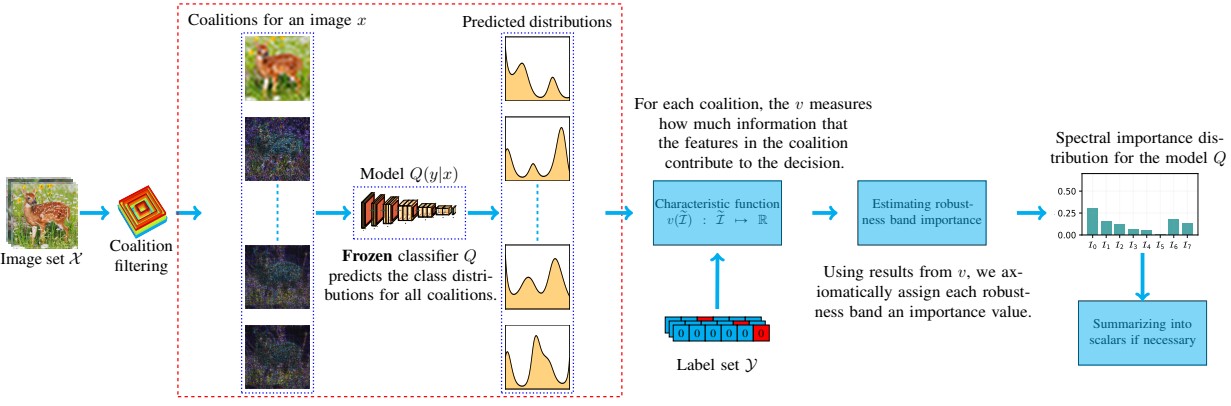

**Figure 4:** Framework of applying Shapley value theory. Spectral coalition filtering creates spectral coalitions over $\mathcal{X}$. Each coalition contains a unique combination of spectral signals, in which some spectral bands are present and others are absent. The coalitions are fed into a classifier $Q$. For each coalition, $Q$ outputs the predictions. The characteristic function $v$ then uses the predictions to estimate the contributions of the features present in the coalitions. The results from $v$ are combined to compute the marginal contributions of spectral bands – *i.e.* the spectral importance distribution of $Q$.

the player $\mathcal{I}_i$. The Shapley value $\psi_i(\mathcal{I}, v)$ is uniquely given by:

$$\psi_i(\mathcal{I}, v) = \sum_{\tilde{\mathcal{I}} \subseteq \mathcal{I} \setminus \mathcal{I}_i} \frac{1}{M} \binom{M-1}{|\tilde{\mathcal{I}}|}^{-1} \left\{ v(\tilde{\mathcal{I}} \cup \{\mathcal{I}_i\}) - v(\tilde{\mathcal{I}}) \right\} \tag{3}$$

where $\frac{1}{M}\binom{M-1}{|\tilde{\mathcal{I}}|}^{-1}$ gives the weight of the player $\mathcal{I}_i$ presenting in the coalition $\tilde{\mathcal{I}}$.

**Spectral importance distribution (SID)**. We use $\Psi(v) := (\psi_i)_{i \in [M]} \in \mathbb{R}^M$ to denote the collection of $\psi_i(\mathcal{I}, v)$ over all players. We min-max normalize $\Psi(v)$ by taking $\bar{\Psi}(v) = \frac{\Psi(v) - \min \Psi(v)}{||\Psi(v) - \min \Psi(v)||_1}$. The reason for normalizing the SIDs is that we want to scalarize the SIDs for numerical comparisons. The $\bar{\Psi}(v)$ is referred to as *spectral importance distribution*. Figure 7 shows examples of the spectral importance distributions of trained and un-trained models.

**Spectral robustness score (SRS)**. We can also summarize spectral importance distributions into scalar values. We refer to the summarized scalar values as *spectral robustness scores*. We use $S(v) : v \mapsto [0, 1]$ to denote the summarizing function.

## 3.2 Spectral coalition filtering

We represent the *presences* and *absences* of the spectral players through the signal *pass-bands* and *stop-bands* using a multi-band-pass digital signal filtering (Oppenheim, 1978; Pei & Tseng, 1998; Steiglitz, 2020), as shown in Figure 5. For example, the example spectral coalition $\{\mathcal{I}_0, \mathcal{I}_2\}$ signifies the signals present only in $\mathcal{I}_0$ and $\mathcal{I}_2$. With the spectral coalition filtering, we are able to evaluate the contributions of the combinations of various spectral features. Figure 6 shows an example of $2^M$ spectral coalitions.

To implement the presences and absences of spectral signals, we define a mask map $\mathbb{T}(\widetilde{\mathcal{I}}) : \widetilde{\mathcal{I}} \mapsto \{0, 1\}^{M \times N}$ on 2D spectrum, where $\widetilde{\mathcal{I}}$ is a spectral coalition and $M \times N$ denotes 2D image dimensions. The mask map is point-wisely defined as:

$$\mathbb{T}(\widetilde{\mathcal{I}})(m, n) = \begin{cases} 1, & \text{if the frequency point (m,n) is present in coalition } \widetilde{\mathcal{I}}, \\ 0, & \text{otherwise} \end{cases} \tag{4}$$

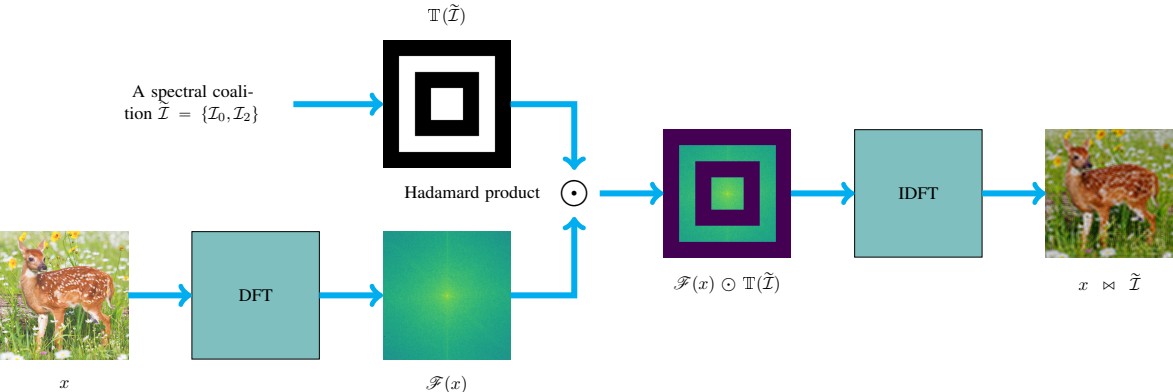

**Figure 5:** Spectral coalition filtering. In this example, the mask map $\mathbb{T}(\widetilde{\mathcal{I}})$ (*i.e.* transfer function) only allows to pass the signals present in the spectral coalition $\{\mathcal{I}_0, \mathcal{I}_2\}$. The $M$ is 4 and the absences are assigned to zeros. The images after spectral coalition filtering ($x \bowtie \widetilde{I}$) are then fed into frozen classifiers to assess the contributions of spectral coalitions. The binary operation notation $\bowtie$ denotes coalition filtering given by Definition 3.1. The notation $\odot$ denotes Hadamard product.

where $(m, n) \in [M] \times [N]$. In the digital signal processing literature, this mask map is known as a *transfer function* (Steiglitz, 2020). In the 2D mask map, the frequency points are '1' in pass-bands (presences) and '0' in stop-bands (absences). The process of spectral coalition filtering with the mask map for a given spectral coalition $\widetilde{\mathcal{I}}$ is shown as in Figure 5.

We apply the transfer function $\mathbb{T}(\widetilde{\mathcal{I}})$ on the spectra of images with element-wise product (*i.e.* Hadamard product (Horn, 1990; Horadam, 2012)). Let $\mathscr{F}$ be the Discrete Fourier transform (DFT) operator and $\mathscr{F}^{-1}$ be the inverse DFT (IDFT) operator (Tan & Jiang, 2018). Readers can further refer to Appendix B.1.

**Definition 3.1** (Spectral coalition filtering). *We define a binary operator '$\bowtie$' to represent the signal filtering by:*

$$x \bowtie \widetilde{\mathcal{I}} := \mathscr{F}^{-1}\left[\underbrace{\mathscr{F}(x) \odot \mathbb{T}(\widetilde{\mathcal{I}})}_{\text{Spectral presence}} + \underbrace{\boldsymbol{b} \odot (\mathbb{1} - \mathbb{T}(\widetilde{\mathcal{I}}))}_{\text{Spectral absence}}\right] \tag{5}$$

*where '$\odot$' denotes Hadamard product (i.e. element-wise product), $\mathbb{1} \in \mathbb{R}^{M \times N}$ denotes an all-ones matrix and $\boldsymbol{b} \in \mathbb{C}^{M \times N}$ represents the assignments of the absences of spectral players. In the context of attribution analysis, $\boldsymbol{b}$ is often referred as the baseline. In our implementation, we empirically set $\boldsymbol{b} = \boldsymbol{0}$.*

**Remark 3.2** (Absence baseline). *Formally, in the context of attribution analysis, the term 'baseline' defines the absence assignments of players (Sundararajan et al., 2017; Shrikumar et al., 2017; Binder et al., 2016). For example, if we use 'zeros' to represent the absence of players, the 'zeros' are dubbed as the 'baseline' in the attribution analysis. We have discussed multiple baselines in Appendix B.2.*

**Definition 3.3** (Spectral coalition filtering over set). *Accordingly, we define the filtering over a set $\mathcal{X}$ as:*

$$\mathcal{X} \bowtie \widetilde{\mathcal{I}} := \{x \bowtie \widetilde{\mathcal{I}} | x \in \mathcal{X}\}. \tag{6}$$

### 3.3 Characteristic function design

The characteristic function is needed in order to measure the contributions of the features in $\widetilde{\mathcal{I}}$. We define the characteristic function as the gains of the negative cross-entropy loss values between feature presences and absences.

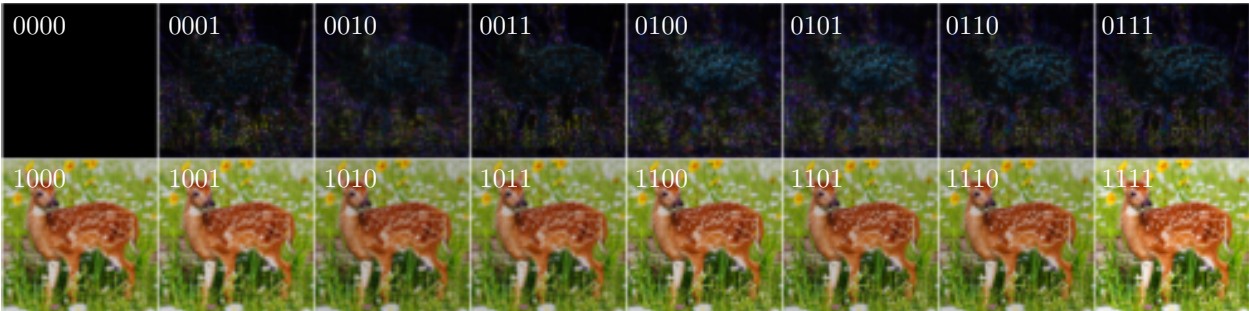

**Figure 6:** An example of a complete $2^M$ spectral coalitions. This example shows 16 spectral coalitions with $M = 4$. Each coalition provides various information relevant to decisions. Each image is a coalition. Each coalition contains a unique spectral signal combination. We use binary code to index these coalitions. The '1' in the $i$-th position indicates the presence of the $i$-th player. For example, 1001 indicates the presences of two spectral players ($\mathcal{I}_0$ and $\mathcal{I}_3$) in the coalition.

**Definition 3.4** (Characteristic function). *The characteristic function $v(\widetilde{\mathcal{I}}) : \widetilde{\mathcal{I}} \mapsto \mathbb{R}$ is defined as:*

$$v(\widetilde{\mathcal{I}}) := \underset{x \sim \mathcal{X}}{\mathbb{E}} \sum_{y \in \mathcal{Y}} \left\{ \underbrace{P(y|x) \cdot \log Q(y|x \bowtie \widetilde{\mathcal{I}})}_{\text{Feature presence}} - \underbrace{P(y|x) \cdot \log Q(y|x \bowtie \varnothing)}_{\text{Absence baseline}} \right\}$$

$$= \underset{x \sim \mathcal{X}}{\mathbb{E}} \sum_{y \in \mathcal{Y}} P(y|x) \cdot \log Q(y|x \bowtie \widetilde{\mathcal{I}}) - C \tag{7}$$

*where the constant term $C := \underset{x \sim \mathcal{X}}{\mathbb{E}} \sum_{y \in \mathcal{Y}} P(y|x) \cdot \log Q(y|x \bowtie \varnothing)$ is used to fulfil $v(\varnothing) = 0$. We refer to the $C$ as the Dummy player constant.*

**Remark 3.5.** *If the labels are one-hot, then Equation 7 is simplified into:*

$$v(\widetilde{\mathcal{I}}) := \underset{x,y \sim \langle \mathcal{X}, \mathcal{Y} \rangle}{\mathbb{E}} \log Q(y|x \bowtie \widetilde{\mathcal{I}}) - C \tag{8}$$

*and $C := \underset{x,y \sim \langle \mathcal{X}, \mathcal{Y} \rangle}{\mathbb{E}} \log Q(y|x \bowtie \varnothing)$.*

**Linking to information theory**. The relationship between information theory and the characteristic function in the form of negative log-likelihood of Bayes classifiers has been discussed in the literature (Covert et al., 2020; Aas et al., 2021; Lundberg & Lee, 2017). Following on from their discussions, we show that the $v$ in Equation 7 profoundly links to information theory in terms of spectral signals. The maximal information of features relevant to labels in $\widetilde{\mathcal{I}}$ is the mutual information $\mathbb{I}(\mathcal{X} \bowtie \widetilde{\mathcal{I}}, \mathcal{Y})$. A classifier $Q$ can merely utilize a proportion of the information. Theorem 3.6 states an information quantity identity regarding the $\mathbb{I}(\mathcal{X} \bowtie \widetilde{\mathcal{I}}, \mathcal{Y})$ and the $v$. The term $D_{KL}[P||Q]$ measures the point-wise (*i.e.* for an image $x$) KL-divergence between the predictions and ground-truth labels. On the basis of the information quantity identity, the $v$ can be interpreted as the information gains between the presences and absences of the features, which the $Q$ utilizes in decisions from the $\widetilde{\mathcal{I}}$. By enumerating all coalitions, the information gains are then combined to compute the marginal information gains of features in decisions via Equation 3.

**Theorem 3.6** (Spectral coalition information identity). *The information quantity relationship is given as:*

$$\mathbb{I}(\mathcal{X} \bowtie \widetilde{\mathcal{I}}, \mathcal{Y}) \equiv \underset{x \in \mathcal{X} \bowtie \widetilde{\mathcal{I}}}{\mathbb{E}} D_{KL}[P(y|x)||Q(y||x)] + H(\mathcal{Y}) + v(\widetilde{\mathcal{I}}) + C \tag{9}$$

*where $v$ is defined in Equation 7 and $H(\mathcal{Y})$ is the Shannon entropy of the label set $\mathcal{Y}$. The proof is provided in Appendix B.3.*

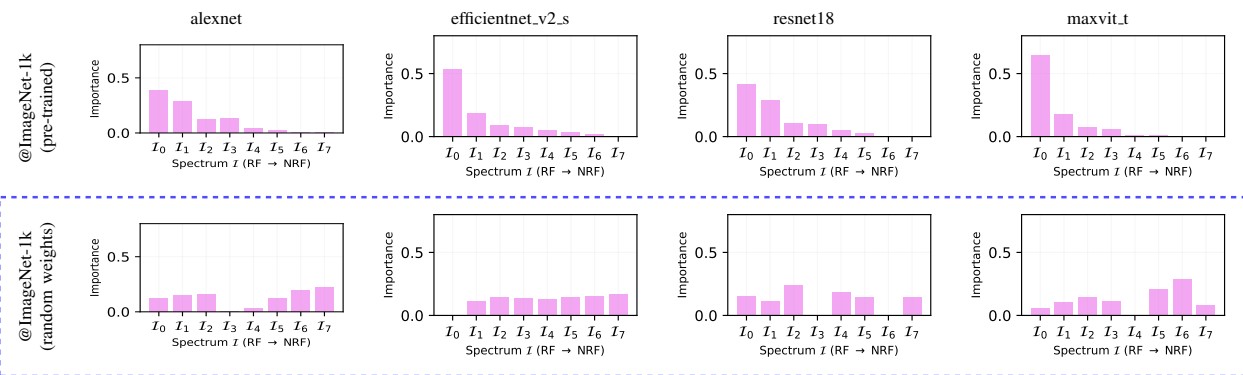

**Figure 7:** Spectral importance distributions (SIDs) of trained models and un-trained models. The experimental models are pre-trained on *ImageNet*. We also include the models with random weights as a control marked by the blue box. We have noticed that: (1) The spectral importance distributions of trained models exhibit non-uniformity, and (2) the spectral importance distributions of un-trained models exhibit uniformity. We summarize these empirical observations as Assumption 3.7.

### 3.4 Spectral robustness score (SRS)

Although we are firstly interested in using the spectral importance distributions (SID) for robustness interpretations, they can also be summarized into scalar scores for purposes such as numerical comparisons, and later correlation studies.

**Assumption 3.7** (Spectral uniformity assumption of random decisions)**.** *The second row in Figure 7 shows the SIDs from various models with randomized weights. We randomize the model weights with Kaiming initialization (He et al., 2015). The measured SIDs exhibit spectral uniformity. This suggests: **Un-trained models do not have spectral preferences**. We refer to 'the models with randomized parameters' as random decisions. Therefore, we assume that the SIDs of random decisions are uniform: $\frac{1}{M}$.*

**Assumption 3.8** (Robustness prior)**.** *We assume: **Higher utilization of robust features in decisions implies robust models**. This is further substantiated by the experiments in Figure 3. To reflect this robustness prior, we empirically design a series $\boldsymbol{\beta} := (\beta^0, \beta^1, \cdots, \beta^{M-1})^T$ where $\beta \in (0, 1)$ as the summing weights of SIDs. Empirically, we choose $\beta = 0.75$ because this choice achieves the best correlation with model robustness.*

**Summarizing with weighted sum**. Let $\Psi(v)$ be the measured spectral importance distribution (SID). Set $\bar{\Psi}(v) = \frac{\Psi(v) - \min \Psi(v)}{||\Psi(v) - \min \Psi(v)||_1}$ with min-max normalization. The weighted sum of the $\Psi(v)$ with the weights $\boldsymbol{\beta}$ is given by:

$$\left| \boldsymbol{\beta}^T \bar{\Psi}(v) - \boldsymbol{\beta}^T \frac{\mathbb{1}}{M} \right| \tag{10}$$

where $\boldsymbol{\beta}^T \frac{\mathbb{1}}{M}$ is served as a random decision baseline. Let $S(v) : v \mapsto [0, 1]$ be the normalized result in Equation 10. The $S(v)$ is given by:

$$S(v) := \frac{\left| \boldsymbol{\beta}^T \bar{\Psi}(v) - \boldsymbol{\beta}^T \frac{\mathbb{1}}{M} \right|}{\sup_{\bar{\Psi}} \left| \boldsymbol{\beta}^T \bar{\Psi}(v) - \boldsymbol{\beta}^T \frac{\mathbb{1}}{M} \right|} = \left| \frac{\bar{\boldsymbol{\beta}}^T \bar{\Psi}(v) - \eta}{1 - \eta} \right| \tag{11}$$

where $\beta \in (0, 1)$, $\bar{\boldsymbol{\beta}} = \frac{\boldsymbol{\beta}}{||\boldsymbol{\beta}||_2}$ and $\eta = \frac{1}{M} \frac{||\boldsymbol{\beta}||_1}{||\boldsymbol{\beta}||_2}$. Readers can refer to Appendix C.2 for the simplification deduction.

## 4 Experiments

We design experiments to show the dual functionality of **I-ASIDE**, which can not only **measure** robustness and but also **interpret** robustness. We organize the experiments in three categories:

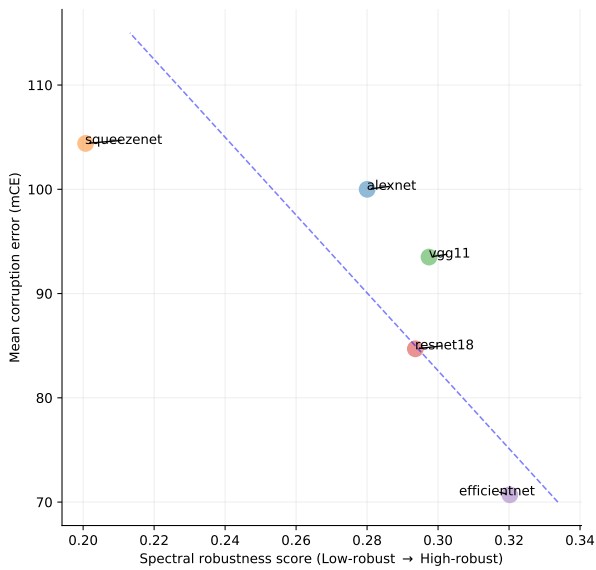

**Figure 8:** The spectral robustness scores (SRS), measured with I-ASIDE, correlate to the mean corruption errors (mCE) in the literature (Hendrycks & Dietterich, 2019).

- Section 4.1: Correlation to a variety of robustness metrics;

- Section 4.2: Studying architectural robustness;

- Section 4.3: A case study interpreting how supervision noise levels affect model robustness.

Section 4.1 shows that the scores obtained with **I-ASIDE** correlate with the model robustness scores measured with other methods. Section 4.2 and Section 4.3 show that **I-ASIDE** is able to interpret robustness by examining SIDs.

**Reproducibility**. We choose $M = 8$ and 200 samples to conduct experiments. We choose 200 samples because the experiments in Appendix C.3. The experiment shows that: **A small amount of examples are sufficiently representative for spectral signals**.

### 4.1 Correlation to robustness metrics

**Definition 4.1** (**Mean prediction error**). *In our experiments, we measure model perturbation robustness with mean prediction errors (mPE) besides the mean corruption errors (mCE). Let x be some clean image and x\* be the perturbed image. For a classifier Q, we define the mean prediction error (mPE) as:*

$$\Delta \mathcal{P} := \mathop{\mathbb{E}}_{x,y\sim\langle\mathcal{X},\mathcal{Y}\rangle} |Q(y|x) - Q(y|x^*)|. \tag{12}$$

We demonstrate that **I-ASIDE** is able to measure model robustness. The experiments are broken down into three aspects: (1) correlation to mCE scores, (2) correlation to adversarial robustness, and (3) correlation to corruption robustness.

**Correlation to mCE scores**. Figure 8 shows the correlation between spectral robustness scores (SRS) and the mean corruption errors (mCE). The mCE scores are taken from the literature (Hendrycks & Dietterich, 2018). The mCE scores are measured on a corrupted *ImageNet* which is known as *ImageNet-C* in the literature (Hendrycks & Dietterich, 2019). The *ImageNet-C* includes 75 common visual corruptions with five levels of severity in each corruption. This correlation suggests that the results measured with **I-ASIDE** correlate with the results measured with robustness metric mCE.

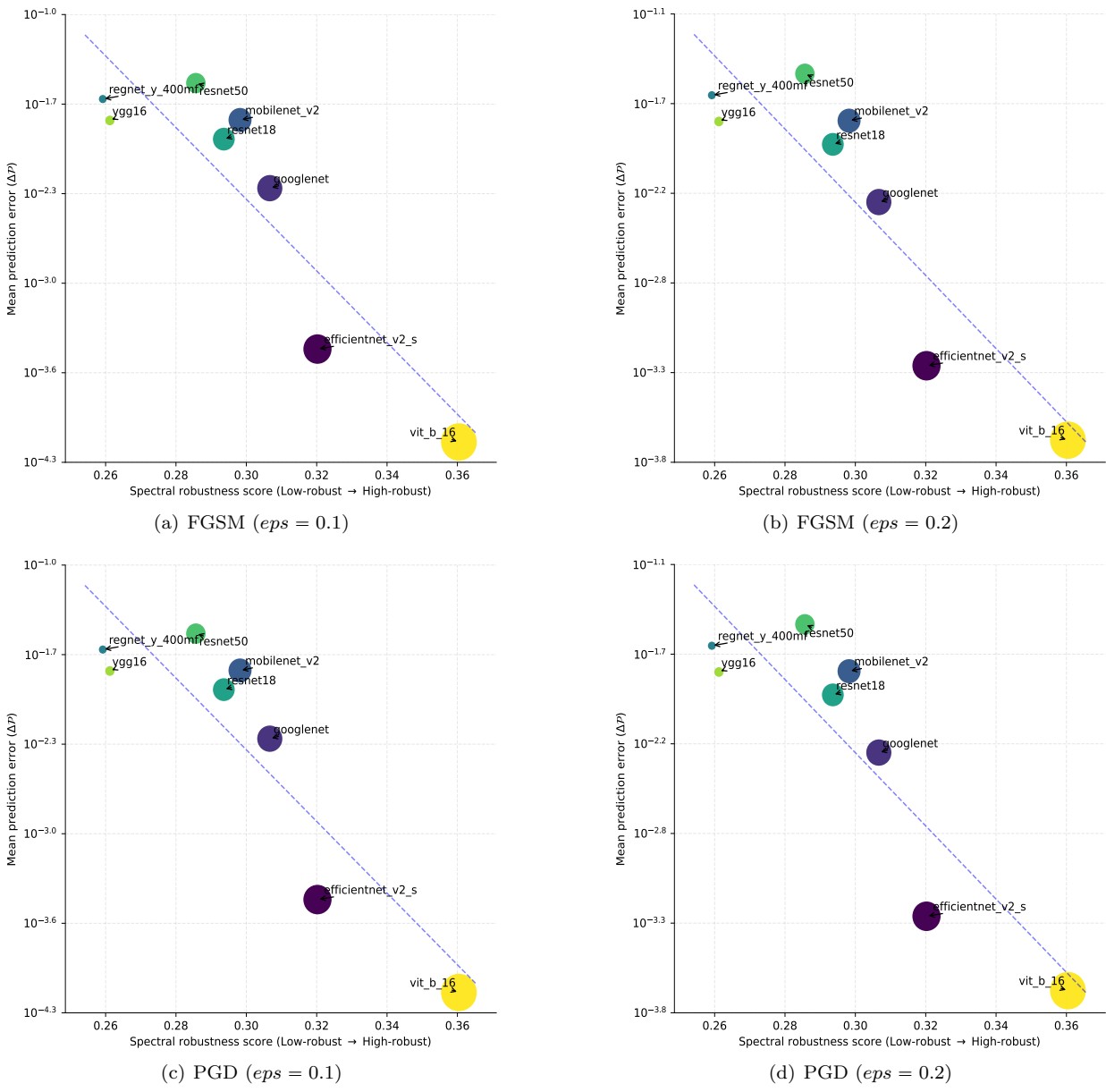

**Figure 9:** The spectral robustness scores (SRS), measured with I-ASIDE, correlate with the mean prediction errors (mPE) in adversarial attacks. The circle sizes in (b) are proportional to the SRS.

**Correlation to adversarial robustness**. Figure 9 shows the correlation between the correlation between spectral robustness scores (SRS) and the mean prediction errors (mPE) of the adversarial attacks with FGSM and PGD. We vary the *eps* from 0.1 to 0.2. The results show that our scores correlate with the mean prediction errors in various *eps* settings. This suggests that the results measured by our method correlate with adversarial robustness.

**Correlation to corruption robustness**. Figure 10 shows the correlation between the correlation between spectral robustness scores (SRS) and the mean prediction errors (mPE) of the corruptions with white noise and Gaussian blurring. We vary the $\sigma$ of white noise from 0.1 to 0.2. We vary the Gaussian blurring kernel sizes from $3 \times 3$ to $7 \times 7$. The results show that our scores correlate with the mean prediction errors in all cases. This suggests that the results measured by our method can reflect the corruption robustness.

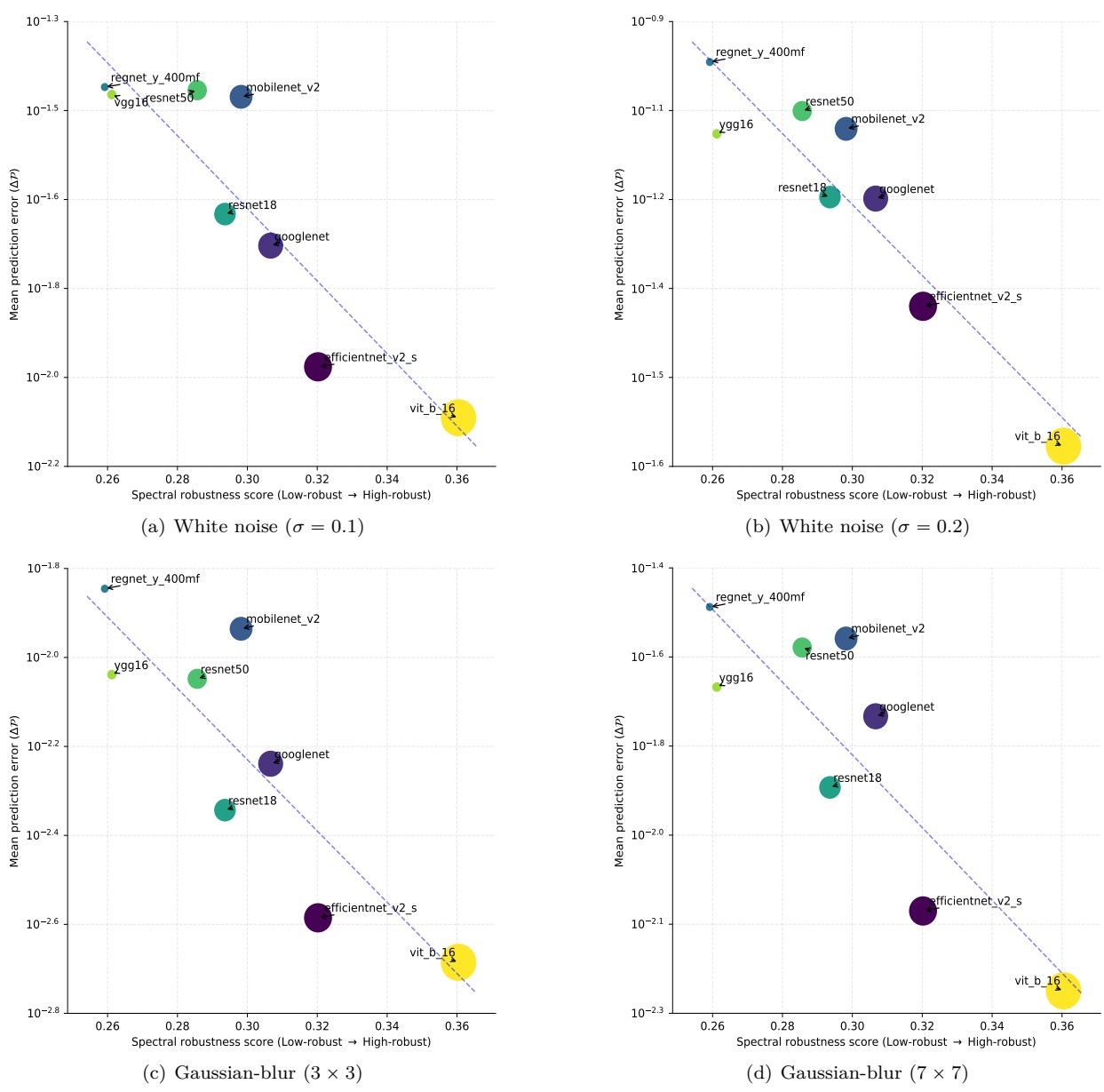

**Figure 10:** The spectral robustness scores (SRS), measured with I-ASIDE, correlate to the mean prediction errors (mPE) in corruptions. The circle sizes in (b) are proportional to the SRS.

## 4.2 Studying architectural robustness

**I-ASIDE** is able to answer questions such as:

- Does model parameter size play a role in robustness?

- Are vision transformers more robust than convolutional neural networks (ConvNets)?

**Does model parameter size play a role in robustness**? Figure 11 (a) shows parameter counts do not correlate with model robustness. Thus, **the tendency of a model to use robust features is not determined by parameter counts alone**. We would like to carry out further investigation in future work.

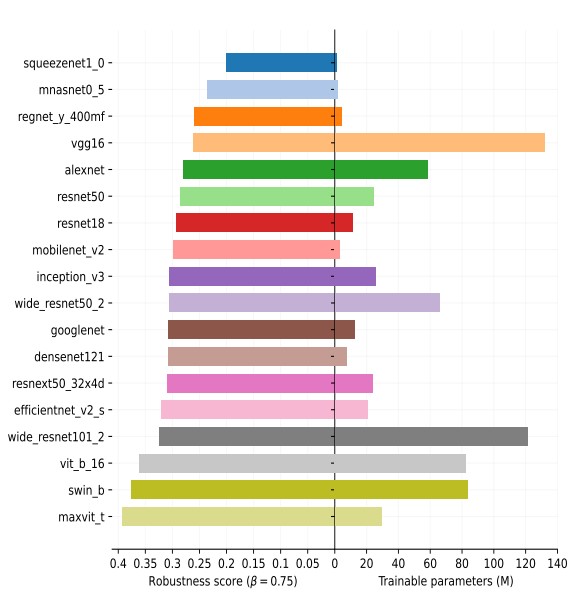

(a) Spectral robustness score (SRS) comparison

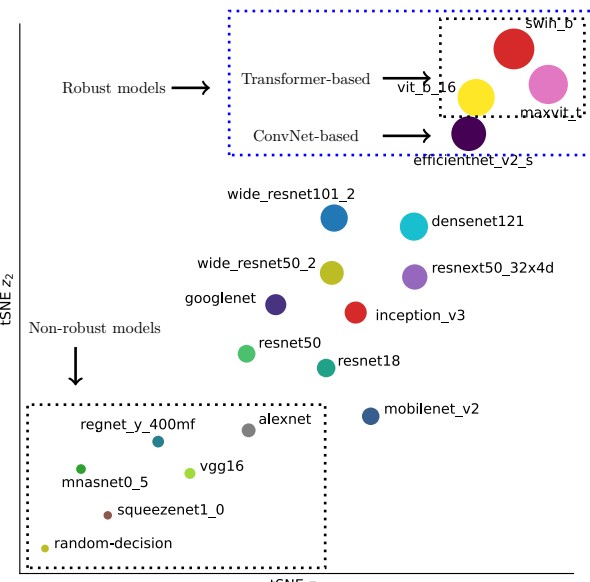

(b) Spectral importance distribution (SID) t-SNE projection

**Figure 11:** How do architectural elements affect robustness? The left figure is to answer: "Does model parameter size play a role on robustness?". The right figure, a t-SNE projection of SIDs, is to answer: "Are vision transformers more robust than convolutional neural networks?". For the two questions, the answers suggested by using **I-ASIDE** are yes. But the story is more complicated, we have provided a brief discussion regarding this case study within Section 4.2. We also have noted that *efficientnet* surprisingly exhibits comparable perturbation robustness as the architectures in Transformer family. All experimental models are pre-trained on *ImageNet*. The circle sizes in (b) are proportional to SRS.

**Are vision transformers more robust than ConvNets**? Figure 11 (b) shows a t-SNE projection of the spectral importance distributions of a variety of models. The results show that vision transformers form a cluster (*swin_b*, *maxvit_t* and *vit_b_16*) and outperform ConvNets in terms of robustness. This results correlate with the recent robustness research in the literature (Paul & Chen, 2022; Zhou et al., 2022; Shao et al., 2021). The interpretation is that: **Vision transformers tend to use more robust features than ConvNets**.

**Discussion**. Vision transformers generally outperform ConvNets; nevertheless, state-of-the-art ConvNets, *e.g. efficientnet* (Tan & Le, 2019), can achieve comparable robustness performance (*e.g.* by error rates on benchmark datasets) (Li & Xu, 2023). The literature (Devaguptapu et al., 2021) affirms that *efficientnet* is more robust than most ConvNets. But, why *efficientnet* is unique? The *efficientnet* introduces an innovative concept in that the network sizes can be controlled by scaling the width, depth, and resolution with a compound coefficient (Tan & Le, 2019). The base architecture is then searched with neural architecture searching (NAS) (Ren et al., 2021) instead of hand-crafted design. The NAS optimization objective is to maximize the network accuracy subject to arbitrary image resolutions. The searching implicitly encourages that the network structure of *efficientnet* uses more robust features. This is because: **The low-frequency signals in various resolutions are robust signals while high-frequency signals are not.** The second column in Figure 7 shows the SID of *efficientnet* pre-trained on *ImageNet*. The SID shows that *efficientnet_v2_s* uses more robust features than *alexnet* and *resnet18*.

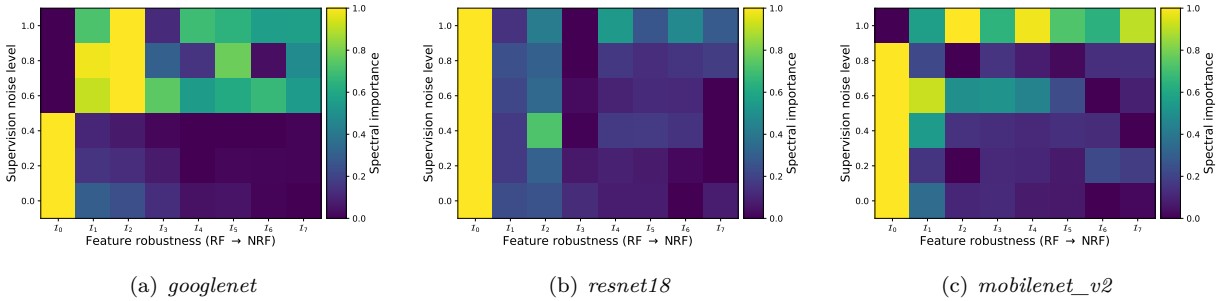

(a) *googlenet*      (b) *resnet18*      (c) *mobilenet_v2*

**Figure 12:** How do models respond to label noise? Our results show that models trained with higher label noise levels tend to use spectral signals uniformly, *i.e.* without a preference for robust (low-frequency) features.

### 4.3 Interpreting how supervision noise levels affect model robustness

The previous robustness benchmarks with mean corruption errors (mCE) are not able to answer the long-standing question: "**How and why label noise levels affect robustness?**". We demonstrate that **I-ASIDE** is able to answer this question.

**Learning with noisy labels**. Supervision signals refer to the prior knowledge provided by labels (Sucholutsky et al., 2023; Zhang et al., 2020; Shorten & Khoshgoftaar, 2019; Xiao et al., 2020). There is a substantial line of previous research on the question of "how supervision noise affects robustness" (Gou et al., 2021; Frénay & Verleysen, 2013; Lukasik et al., 2020; Rolnick et al., 2017). This question is not completely answered yet. For example, Flatow & Penner add uniform label noise into *CIFAR-10* and study its impact on model robustness (Flatow & Penner, 2017). Their results show that classification test accuracy decreases as the training label noise level increases. However, empirical studies like this are not able to answer the underlying 'why' question.

**Noisy-label dataset**. We derive noisy-label datasets from a clean *Caltech101*. We randomly assign a proportion of labels with a uniform distribution over label classes to create a noisy-label dataset. We refer to the randomly assigned proportion as supervision noise level. We vary the noise level from 0.2 to 1.0 to derive five training datasets.

**Experiment**. We train three models (*googlenet*, *resnet18* and *mobilenet_v2*) over the clean and the five noisy-label datasets for 120 epochs respectively. We then measure their SIDs. The results are visualized in Figure 12 with heat maps. The results show that there is a pattern across the above three models in that: **The SIDs are more uniform with higher supervision noise levels**. The interpretation regarding the learning dynamics with the presence of label noise is that: **Models tend to use more non-robust features in the presence of higher label noise within training set**.

## 5 Related work

We further conduct a literature investigation from three research lines: (1) global interpretability, (2) model robustness, and (3) frequency-domain research. This literature study shows that **I-ASIDE** provides unique insights in these research lines.

**Global interpretability**. Global interpretability summarizes the decision behaviours of models from a holistic view. In contrast, local interpretability merely provides explanations on the basis of instances (Sundararajan et al., 2017; Smilkov et al., 2017; Linardatos et al., 2020; Selvaraju et al., 2017; Arrieta et al., 2020; Zhou et al., 2016; Ribeiro et al., 2016; Lundberg & Lee, 2017; Lakkaraju et al., 2019; Guidotti et al., 2018; Bach et al., 2015; Montavon et al., 2019; Shrikumar et al., 2017). There are four major research lines in image models: (1) feature visualization, (2) network dissection, (3) concept-based method, and (4) feature importance.

Feature visualization seeks the ideal inputs for specific neurons or classes by maximizing activations (Olah et al., 2017; Nguyen et al., 2019; Zeiler et al., 2010; Simonyan et al., 2013; Nguyen et al., 2016a;b). This method provides intuitions regarding the question: "What inputs maximize the activations of specific neurons or classes?". Network dissection aims to connect the functions of network units (*e.g. channels* or *layers*) with specific concepts (*e.g. eyes* or *ears*) (Bau et al., 2017). Concept-based methods understand the decisions by answering the question "how do models use a set of given concepts in decisions?" (Kim et al., 2018; Ghorbani et al., 2019; Koh et al., 2020; Chen et al., 2020). For example, TCAV explains model decisions by evaluating the importance of a given set of concepts (*e.g.* the textures *dotted*, *striped* and *zigzagged*) for a given class (*e.g.* the class *zebra*) (Kim et al., 2018).

Global input feature importance analysis, often by using Shapley value theory framework, attempts to answer the question: "How do input features contribute to predictions?" (Altmann et al., 2010; Greenwell et al., 2018; Lundberg & Lee, 2017; Ribeiro et al., 2016; Simonyan et al., 2013; Sundararajan et al., 2017; Covert et al., 2020). However, there are few works falling in the scope of global interpretability with feature importance analysis. A related work, SAGE, applying the Shapley value theory framework, globally assigns spatial input features with importance values for interpreting spatial feature contributions (Covert et al., 2020).

Although the aforementioned global interpretability methods provide insights into understanding decisions inside black-box models, they do not provide interpretations regarding robustness mechanisms. Our work fundamentally differs from them in that: We provide interpretations regarding robustness mechanisms. We attempt to answer the fundamental question: "Why some models are more robust than others?".

**Model robustness**. Model robustness refers to the prediction sensitivity of models to perturbations. The perturbations can perturb in spaces such as the input space and the parameter space (Hendrycks & Dietterich, 2019; Drenkow et al., 2021). In this research, we focus on the perturbations within the input space. The perturbations can stem from sources such as adversarial attacks (Szegedy et al., 2013; Goodfellow et al., 2014), corruptions (Hendrycks & Dietterich, 2019), outliers (Hendrycks et al., 2018a; Pang et al., 2021) and supervision signal noise (Hendrycks et al., 2018b).

Model robustness is often assessed using scalar metrics (Hendrycks & Dietterich, 2019; Krizhevsky et al., 2012; Laugros et al., 2019; Taori et al., 2020). For example, robustness can be measured by the distances between clean and perturbed pairs in feature spaces (Zheng et al., 2016). Hendrycks & Dietterich benchmark the corruption robustness with mean corruption errors (mCE) over a set of corrupted datasets like *ImageNet-C* (Hendrycks & Dietterich, 2019), using AlexNet (Hendrycks & Dietterich, 2019) as a normalization baseline.

Despite their widespread adoption in previous literature, these scalar metrics lack the ability to provide detailed insights into the robustness mechanisms. Our work not only serves as a robustness metric but also offers mechanistic interpretations, answering the "why" question behind model robustness. This dual functionality distinguishes our approach, providing a deeper understanding of the mechanisms.

**Frequency-domain research**. Neural networks are non-linear parameterized signal processing filters. Investigating how neural networks respond to input signals in the frequency-domain can provide a unique insight into understanding its functions. Xu et al. delve into the learning dynamics of neural networks in the frequency-domain (Xu et al., 2019a;b). They present their findings as 'F-Principle'. Their work suggests that the learning behaviors of neural networks exhibit spectral non-uniformity: Neural networks fit low-frequency components first, then high-frequency components.

In a related study, Tsuzuku & Sato show that convolutional neural networks have spectral non-uniformity with respect to Fourier bases (Tsuzuku & Sato, 2019). Later, Wang et al. connect model generalization behaviors and image spectrum (Wang et al., 2020). They argue that: (1) The supervision signals provided by humans use more low-frequency signals in images and (2) models trained on it tend to use more low-frequency signals. Our showcase experiment in Figure 12 provides the interpretations regarding their empirical findings.

In the interpretability research line within the frequency-domain, Kolek et al. propose 'CartoonX' based on the rate-distortion explanation (RDE) framework (Macdonald et al., 2019; Heiß et al., 2020). The RDE framework identifies decision-critical features by partially obfuscating the features. They refer to 'the prediction errors between clean inputs and the partially obfuscated inputs' as distortions. CartoonX pinpoints the decision-critical features within wavelet domain to answer the query: "What features are crucial for

decisions?" (Kolek et al., 2022). Our work differs from CartoonX in that: (1) Our method aims to interpret model robustness mechanisms while CartoonX does not, (2) our method is a global interpretability method while CartoonX is a local interpretability method, (3) our method analyzes within an information theory framework while CartoonX uses RDE framework, and (4) our method uses Fourier bases while CartoonX uses wavelet bases.

## 6  Limitations

**I-ASIDE** provides a unique insight into the perturbation robustness mechanisms. Yet, our method has two major limitations: (1) The spectral perspective can merely reflect one aspect of the holistic view of model robustness, and (2) the SID resolutions are low.

**Limitation (1)**. For example, carefully crafted malicious adversarial perturbations on low-frequency components can fool neural networks (Luo et al., 2022; Liu et al., 2023; Maiya et al., 2021). Luo et al. demonstrate that attacking low-frequency signals can fool neural networks, resulting in attacks which are imperceptible to humans. This further implies the complexity of this research topic.

**Limitation (2)**. The computation cost is imposed by $\mathcal{O}(2^M)$. Fortunately, we do not need high SID resolution to analyze the model robustness problem. For example, a choice with $M = 8$ is sufficient to interpret robustness mechanisms (as we have shown) while the computational cost remains reasonable.

## 7  Conclusions

On the solid ground provided by information theory and coalitional game theory, we present an axiomatic method to interpret model robustness mechanisms, by leveraging the power-law-like decay of SNRs over the frequency. Our method addresses the limitation that scalar metrics fail to interpret robustness mechanisms. We carry out extensive experiments over a variety of architectures. The SIDs, when scalarized, can largely reproduce the results found with previous methods, but addresses their failures to answer the underlying 'why' questions. Our method goes beyond them with the dual functionality in that: **I-ASIDE** can not only measure the robustness but also interpret its mechanisms. Our work provides a unique insight into the robustness mechanisms of image classifiers.

## Acknowledgments

This publication has emanated from research [conducted with the financial support of/supported in part by a grant from] Science Foundation Ireland under Grant number 18/CRT/6223. For the purpose of Open Access, the author has applied a CC BY public copyright licence to any Author Accepted Manuscript version arising from this submission. We also thank reviewers for their constructive comments which can significantly improve our research quality. We thank the support from the ICHEC (Irish Centre for High-End Computing). We also thank the help from Prof. Dr. Michael Madden from University of Galway, Ireland.

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

## A  Appendix

## B  Fairness division axioms

***Symmetry* axiom**: Let $\widetilde{\mathcal{I}} \in 2^{\mathcal{I}}$ be some spectral player coalition. For $\forall \ \mathcal{I}_i, \mathcal{I}_j \in \mathcal{I} \wedge \mathcal{I}_i, \mathcal{I}_j \notin \widetilde{\mathcal{I}}$, the statement $v(\widetilde{\mathcal{I}} \cup \{\mathcal{I}_i\}) = v(\widetilde{\mathcal{I}} \cup \{\mathcal{I}_j\})$ implies $\psi_i(\mathcal{I}, v) = \psi_j(\mathcal{I}, v)$. This axiom restates the statement '*equal treatment of equals*' principle mathematically. This axiom states that the 'names' of players should have no effect on the 'treatments' by the characteristic function in coalition games (Roth, 1988).

***Linearity* axiom**: Let $u$ and $v$ be two characteristic functions. Let $(\mathcal{I}, u)$ and $(\mathcal{I}, v)$ be two coalition games. Let $(u + v)(\widetilde{\mathcal{I}}) := u(\widetilde{\mathcal{I}}) + v(\widetilde{\mathcal{I}})$ where $\widetilde{\mathcal{I}} \in 2^{\mathcal{I}}$. The divisions of the new coalition game $(\mathcal{I}, u + v)$ should satisfy: $\psi_i(\mathcal{I}, u + v) = \psi_i(\mathcal{I}, u) + \psi_i(\mathcal{I}, v)$. This axiom is also known as '*additivity* axiom' and guarantees the uniqueness of the solution of dividing payoffs among players (Roth, 1988).

***Efficiency* axiom**: This axiom states that the sum of the divisions of all players must be summed to the worth of the player set (the grand coalition): $\sum_{i=0}^{M-1} \psi_i(\mathcal{I}, v) = v(\mathcal{I})$.

***Dummy player* axiom**: A dummy player (null player) $\mathcal{I}_*$ is the player who has no contribution such that: $\psi_*(\mathcal{I}, v) = 0$ and $v(\widetilde{\mathcal{I}} \cup \{\mathcal{I}_*\}) \equiv v(\widetilde{\mathcal{I}})$ for $\forall \ \mathcal{I}_* \notin \widetilde{\mathcal{I}} \wedge \mathcal{I}_* \subseteq \mathcal{I}$.

**Remark B.1.** *In the literature (Roth, 1988), the efficiency axiom and the dummy player axiom are also combined and relabeled as carrier axiom.*

### B.1 Spectral signal-to-noise ratio (SNR)

**Discrete Fourier Transform**. The notion 'frequency' measures how 'fast' the outputs can change with respect to inputs. High frequency implies that small variations in inputs can cause large changes in outputs. In terms of images, the 'inputs' are the pixel spatial locations while the 'outputs' are the pixel values.

Let $x : (i, j) \mapsto \mathbb{R}$ be some 2D image with dimension $M \times N$ which sends every location $(i, j)$ to some real pixel value where $(i, j) \in [M] \times [N]$. Let $\mathscr{F} : \mathbb{R}^2 \mapsto \mathbb{C}^2$ be some DFT functional operator. The DFT of $x$ is given by:

$$\mathscr{F}(x)(u, v) = \sum_{j=0}^{N-1} \sum_{i=0}^{M-1} x(i, j) e^{-\mathbf{i} 2\pi (\frac{u}{M} i + \frac{v}{N} j)}. \tag{13}$$

**Point-wise energy spectral density (ESD)**. The ESD measures the energy quantity at a frequency. To simplify discussions, we use *radial frequency*, which is defined as the radius $r$ with respect to zero frequency point (*i.e.* the frequency center). The energy is defined as the square of the frequency magnitude according to Parseval's Power Theorem.

Let $L_r$ be a circle with radius $r$ on the spectrum of image $x$, as illustrated in Figure 1. The $r$ is referred to as radial frequency. The point-wise ESD function is given by:

$$ESD_r(x) := \frac{1}{|L_r|} \cdot \sum_{(u,v) \in L_r} |\mathscr{F}(x)(u, v)|^2 \tag{14}$$

where $(u, v)$ is the spatial frequency point and $|L_r|$ is the circumference of $L_r$.

**Spectral signal-to-noise ratio (SNR)**. The SNR can quantify signal robustness. We define the spectral SNR at radius frequency $r$ as:

$$SNR(r) := \frac{ESD_r(x)}{ESD_r(\Delta x)} \tag{15}$$

where $\Delta x$ is some perturbation. We have characterized the SNRs of some corroptions and adversarial attacks in Figure 2.

## B.2 Absence assignment scheme

There exist multiple choices for the assignments of the absences of spectral layers in coalition filtering design: (1) Assigning to constant zeros (Zeroing), (2) assigning to complex Gaussian noise (Complex Gaussian) and (3) assigning to the corresponding frequency components randomly sampled from other images at the same dataset (Replacement).

**Zeroing**. The $\boldsymbol{b}$ in Equation 5 is set to zeros.

**Complex Gaussian**. The $\boldsymbol{b}$ in Equation 5 is sampled from a *i.i.d.* complex Gaussian distribution: $\mathcal{N}(\mu, \frac{\sigma^2}{2}) + i\mathcal{N}(\mu, \frac{\sigma^2}{2})$.

**Replacement**. The $\boldsymbol{b}$ in Equation 5 is set to: $\boldsymbol{b} = \mathscr{F}(x^*)$ (where $x^* \sim \mathcal{X}$ is a randomly sampled image from some set $\mathcal{X}$).

In our implementation, we simply choose 'zeroing': $\boldsymbol{b} = \boldsymbol{0}$. Figure 13 shows the filtered image examples by using the above three strategies and also show the examples of measured spectral importance distributions. Empirically, the three strategies have rather similar performance. In this research, we do not unfold the discussions regarding the masking strategy choices.

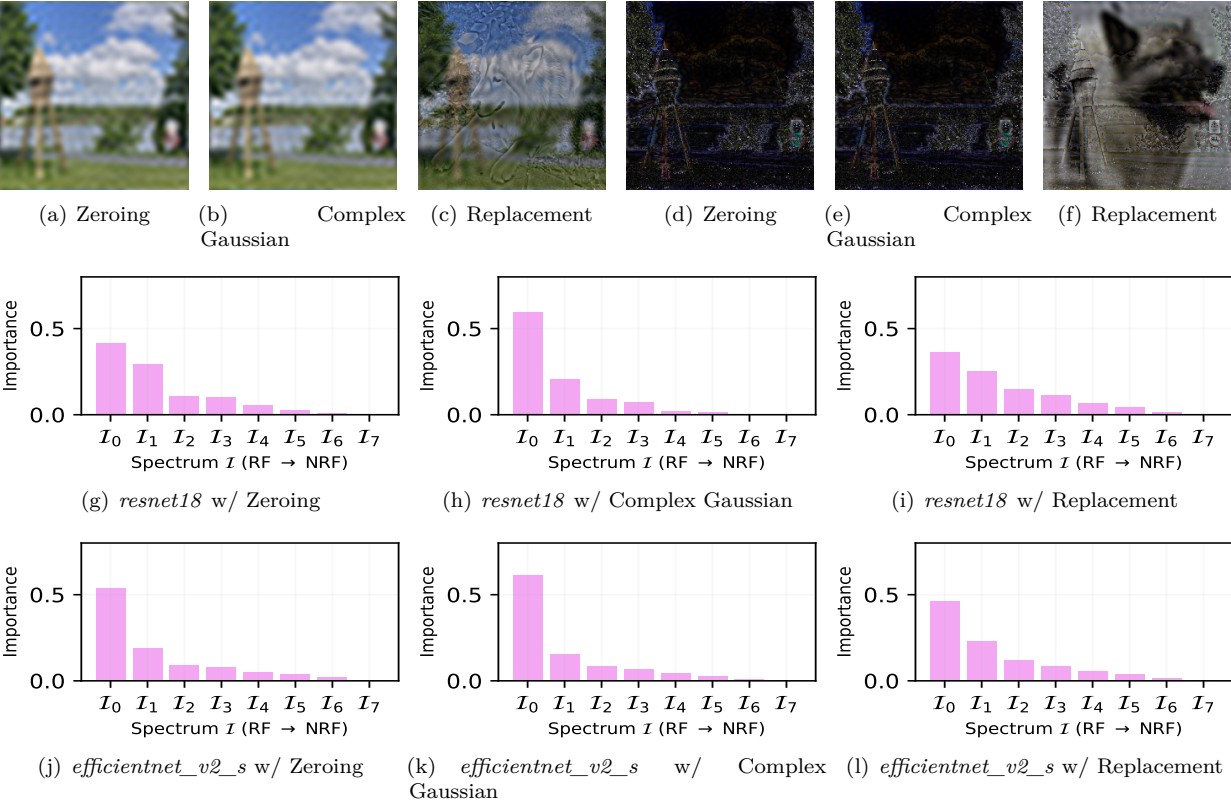

**Figure 13:** Three absence assignment strategies: (1) Assigning the spectral absences with constant zeros (Zeroing), (2) assigning the spevtral absences with Gaussian noise (Complex Gaussian) and (3) randomly sampling spectral components from the same image datasets (Replacement). The standard complex Gaussian distribution is given by: $\mathcal{N}(0, \frac{1}{2}) + i\mathcal{N}(0, \frac{1}{2})$. The figures (a), (b) and (c) show the coalition filtering results with the spectral coalition: $\{\mathcal{I}_0\}$. The figures (d), (e) and (f) show the coalition filtering results with the spectral coalition: $\{\mathcal{I}_1, \mathcal{I}_2, \mathcal{I}_3, \mathcal{I}_4, \mathcal{I}_5, \mathcal{I}_6, \mathcal{I}_7\}$. The figures (g) to (l) show the examples of the measured spectral importance distributions of a *resnet18* and a *efficientnet_v2_s* (both are pre-trained on *ImageNet*) with the three assignment strategies.

### B.3 Proof for Spectral Coalition Information Identity Theorem

*Proof for Spectral Coalition Information Identity.* Suppose the probability measures $P(x)$, $P(x,y)$, $P(y|x)$, and $Q(y|x)$ are absolutely continuous with respect to $x$ on domain $\mathcal{X} \bowtie \widetilde{\mathcal{I}}$.

$$\mathbb{I}(\mathcal{X} \bowtie \widetilde{\mathcal{I}}, \mathcal{Y}) = \int_{\mathcal{X} \bowtie \widetilde{\mathcal{I}}} \sum_{y \in \mathcal{Y}} P(x,y) \cdot \log \frac{P(x,y)}{P(x) \cdot P(y)} dx \tag{16}$$

$$= \int_{\mathcal{X} \bowtie \widetilde{\mathcal{I}}} \sum_{y \in \mathcal{Y}} P(x,y) \cdot \log \left( \frac{P(y|x) \cdot P(x)}{P(y) \cdot P(x)} \cdot \frac{Q(y|x)}{Q(y|x)} \right) dx \tag{17}$$

$$= \int_{\mathcal{X} \bowtie \widetilde{\mathcal{I}}} \sum_{y \in \mathcal{Y}} P(x,y) \cdot \log \left( \frac{P(y|x)}{Q(y|x)} \cdot \frac{1}{P(y)} \cdot Q(y|x) \right) dx \tag{18}$$

$$= \int_{\mathcal{X} \bowtie \widetilde{\mathcal{I}}} P(x) \left( \sum_{y \in \mathcal{Y}} P(y|x) \cdot \log \frac{P(y|x)}{Q(y|x)} \right) dx \tag{19}$$

$$- \sum_{y \in \mathcal{Y}} \left( \int_{\mathcal{X} \bowtie \widetilde{\mathcal{I}}} P(x,y) dx \right) \log P(y) \tag{20}$$

$$+ \int_{\mathcal{X} \bowtie \widetilde{\mathcal{I}}} \sum_{y \in \mathcal{Y}} P(x,y) \cdot \log Q(y|x) dx \tag{21}$$

$$= \underset{x \in \mathcal{X} \bowtie \widetilde{\mathcal{I}}}{\mathbb{E}} \underbrace{KL(P(y|x)||Q(y||x))}_{\text{point-wise}} + H(\mathcal{Y}) + \int_{\mathcal{X} \bowtie \widetilde{\mathcal{I}}} P(x) \left( \sum_{y \in \mathcal{Y}} P(y|x) \cdot \log Q(y|x) \right) dx \tag{22}$$

$$= \underset{x \in \mathcal{X} \bowtie \widetilde{\mathcal{I}}}{\mathbb{E}} \underbrace{KL(P(y|x)||Q(y||x))}_{\text{point-wise}} + H(\mathcal{Y}) + \underset{x \in \mathcal{X} \bowtie \widetilde{\mathcal{I}}}{\mathbb{E}} \sum_{y \in \mathcal{Y}} P(y|x) \cdot \log Q(y|x) \tag{23}$$

$$= \underset{x \in \mathcal{X} \bowtie \widetilde{\mathcal{I}}}{\mathbb{E}} \underbrace{KL(P(y|x)||Q(y||x))}_{\text{point-wise}} + H(\mathcal{Y}) + v(\widetilde{\mathcal{I}}) + C \tag{24}$$

where $H(\mathcal{Y})$ is the Shannon entropy of the label set $\mathcal{Y}$.

$\square$

## C   Information quantity relationship in spectral coalitions

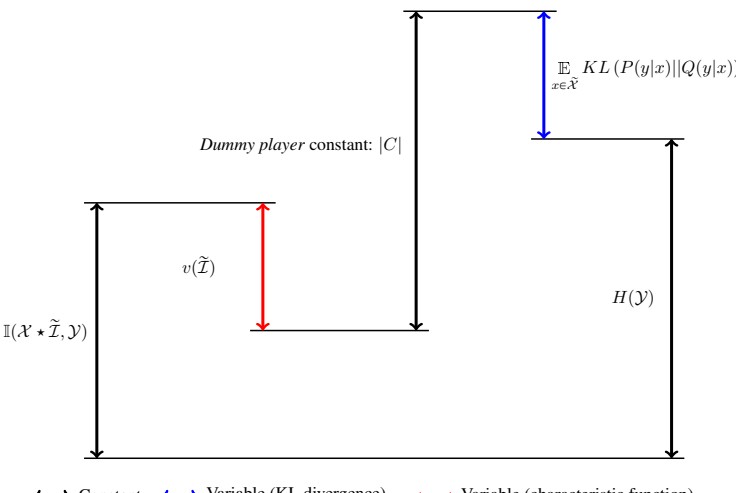

**Figure 14:** Information quantity relationship. This shows the theoretical information quantity relationship between what the characteristic function $v$ measures and the mutual information $\mathbb{I}(\mathcal{X} \bowtie \widetilde{\mathcal{I}}, \mathcal{Y})$. For a given coalition $\widetilde{\mathcal{I}}$, a dataset $\langle \mathcal{X}, \mathcal{Y} \rangle$ and a classifier $Q$, the $v$ measures how much information the classifier $Q$ utilizes in decisions. The measured results are then used to compute the marginal contributions of features.

### C.1 Partitioning spectrum with $\ell_\infty$ ball over $\ell_2$ ball

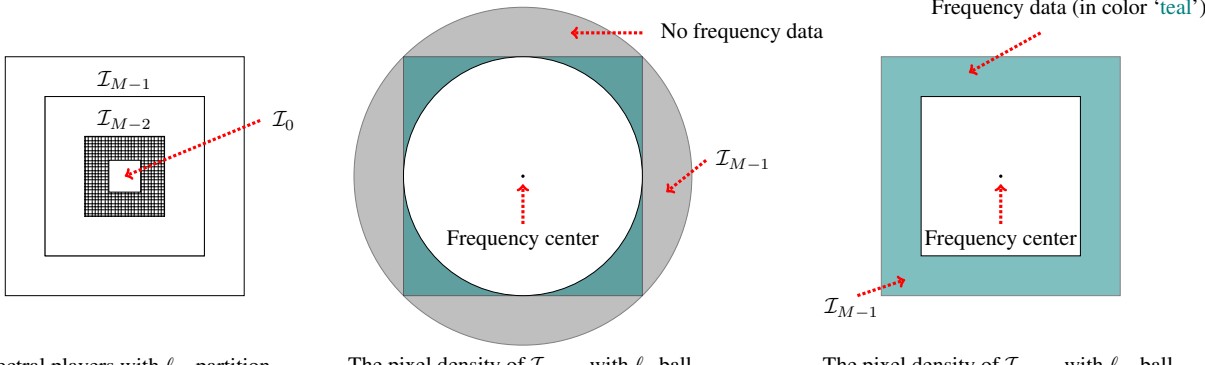

| Spectral players with $\ell_\infty$ partition. | The pixel density of $\mathcal{I}_{M-1}$ with $\ell_2$ ball. | The pixel density of $\mathcal{I}_{M-1}$ with $\ell_\infty$ ball. |

**Figure 15:** Two spectral band partitioning schemes. This shows the motivation we choose $\ell_\infty$ ball over $\ell_2$ ball in partitioning the frequency domain into the $M$ bands (i.e., $M$ 'spectral players') over 2D Fourier spectrum. The frequency data density of the spectral players with $\ell_\infty$ remains a constant. However, the frequency data density of the spectral players with $\ell_2$ is not a constant since some frequency components do not present. This motives us to empirically choose $\ell_\infty$ metric to form spectral players in implementation.

### C.2 Normalizing summarized SIDs

We normalize the above result and set:

$$S(v) := \frac{\left| \boldsymbol{\beta}^T \bar{\Psi}(v) - \frac{||\boldsymbol{\beta}||_1}{M} \right|}{\sup \left| \boldsymbol{\beta}^T \bar{\Psi}(v) - \frac{||\boldsymbol{\beta}||_1}{M} \right|} \tag{25}$$

$$= \frac{\left| \boldsymbol{\beta}^T \bar{\Psi}(v) - \frac{||\boldsymbol{\beta}||_1}{M} \right|}{\sup \left| ||\boldsymbol{\beta}||_2 \cdot ||\bar{\Psi}(v)||_2 - \frac{||\boldsymbol{\beta}||_1}{M} \right|} \tag{26}$$

$$= \frac{\left| \boldsymbol{\beta}^T \bar{\Psi}(v) - \frac{||\boldsymbol{\beta}||_1}{M} \right|}{\left| ||\boldsymbol{\beta}||_2 - \frac{||\boldsymbol{\beta}||_1}{M} \right|} \tag{27}$$

$$= \left| \frac{\bar{\boldsymbol{\beta}}^T \bar{\Psi}(v) - \frac{1}{M} \frac{||\boldsymbol{\beta}||_1}{||\boldsymbol{\beta}||_2}}{1 - \frac{1}{M} \frac{||\boldsymbol{\beta}||_1}{||\boldsymbol{\beta}||_2}} \right|. \tag{28}$$

where $\bar{\boldsymbol{\beta}} = \frac{\boldsymbol{\beta}}{||\boldsymbol{\beta}||_2}$ and $\sup \left| \boldsymbol{\beta}^T \bar{\Psi}(v) - \frac{||\boldsymbol{\beta}||_1}{M} \right|$ is derived by:

$$\sup \left| \boldsymbol{\beta}^T \bar{\Psi}(v) - \frac{||\boldsymbol{\beta}||_1}{M} \right| = \left| \sup \boldsymbol{\beta}^T \bar{\Psi}(v) - \frac{||\boldsymbol{\beta}||_1}{M} \right| \tag{29}$$

$$= \left| \sup ||\boldsymbol{\beta}||_2 \cdot ||\bar{\Psi}(v)||_2 - \frac{||\boldsymbol{\beta}||_1}{M} \right| \quad \text{s.t. } ||\bar{\Psi}(v)||_1 = 1 \tag{30}$$

$$= \left| ||\boldsymbol{\beta}||_2 - \frac{||\boldsymbol{\beta}||_1}{M} \right| \quad \text{since } ||\bar{\Psi}(v)||_2^2 \leqslant ||\bar{\Psi}(v)||_1^2. \tag{31}$$

Set $\eta = \frac{1}{M} \frac{||\boldsymbol{\beta}||_1}{||\boldsymbol{\beta}||_2}$:

$$S(v) = \left| \frac{\bar{\boldsymbol{\beta}}^T \bar{\Psi}(v) - \eta}{1 - \eta} \right|. \tag{32}$$

$$Q.E.D.$$

## C.3 How much samples are sufficient?

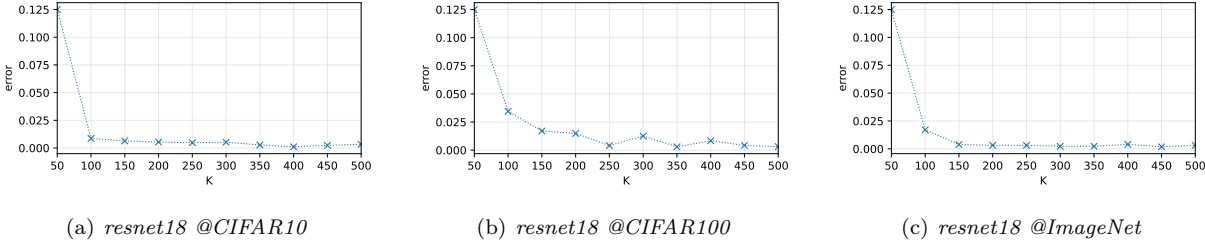

(a) *resnet18 @CIFAR10*  (b) *resnet18 @CIFAR100*  (c) *resnet18 @ImageNet*

**Figure 16:** Convergence of relative estimation errors converge with respect to the numbers of samples $K$. The errors are measured by: $\frac{1}{M}||\Psi^{(i+1)}(v) - \Psi^{(i)}(v)||_1$ where $\Psi^{(i)}(v)$ denotes the $i$-th measured spectral importance distribution with respect to characteristic function $v$. The experiments are conducted on *CIFAR10*, *CIFAR100* and *ImageNet* with *resnet18*.

**Error bound analysis**. Let $K$ be the number of the samples of some baseline dataset. Let:

$$\Delta v(\tilde{\mathcal{I}}, \mathcal{I}_i) := v(\tilde{\mathcal{I}} \cup \{\mathcal{I}_i\}) - v(\tilde{\mathcal{I}}) \tag{33}$$

and

$$\Delta v(\mathcal{I}_i) := \big(\Delta v(\tilde{\mathcal{I}}, \mathcal{I}_i)\big)_{\tilde{\mathcal{I}} \subseteq \mathcal{I}} \tag{34}$$

and

$$W := \left( \frac{1}{M} \binom{M-1}{|\tilde{\mathcal{I}}|}^{-1} \right)_{\tilde{\mathcal{I}} \subseteq \mathcal{I}}. \tag{35}$$

Hence:

$$\psi_i(\mathcal{I}, v) = W^T \Delta v(\mathcal{I}_i) \tag{36}$$

where $||W||_1 \equiv 1$ since $W$ is a probability distribution. Let $\bar{\psi}_i$, $\Delta \bar{v}(\mathcal{I}_i)$ and $\Delta \bar{v}(\tilde{\mathcal{I}}, \mathcal{I}_i)$ be estimations with $K$ samples using Monte Carlo sampling. The error bound with $\ell_1$ norm is given by:

$$\epsilon \stackrel{\text{def}}{=} \sup_i ||\bar{\psi}_i(\mathcal{I}, v) - \psi_i(\mathcal{I}, v)||_1 = \sup_i ||W^T \Delta \bar{v}(\mathcal{I}_i) - W^T \Delta v(\mathcal{I}_i)||_1 \tag{37}$$

$$\leqslant \sup_i ||W||_1 \cdot ||\Delta \bar{v}(\mathcal{I}_i) - \Delta v(\mathcal{I}_i)||_\infty \qquad \big(\text{Hölder's inequality}\big) \tag{38}$$

$$= \sup_i || \sum_{\tilde{\mathcal{I}} \subseteq \mathcal{I} \setminus \mathcal{I}_i} \big(\Delta \bar{v}(\tilde{\mathcal{I}}, \mathcal{I}_i) - \Delta v(\tilde{\mathcal{I}}, \mathcal{I}_i)\big) ||_\infty \tag{39}$$

$$\leqslant \sup_i 2^{M-1} \cdot \sup_{\tilde{\mathcal{I}}} ||\Delta \bar{v}(\tilde{\mathcal{I}}, \mathcal{I}_i) - \Delta v(\tilde{\mathcal{I}}, \mathcal{I}_i)||_\infty \tag{40}$$

$$= \sup_i 2^{M-1} \cdot \sup_{\tilde{\mathcal{I}}} ||\Delta \bar{v}(\tilde{\mathcal{I}}, \mathcal{I}_i) - \Delta v(\tilde{\mathcal{I}}, \mathcal{I}_i)||_1 \tag{41}$$

$$\leqslant 2^{M-1} \cdot \left\{ \frac{Var(\Delta \bar{v})}{K} \right\}^{\frac{1}{2}} \tag{42}$$

where $Var(\Delta \bar{v})$ gives the upper bound of the variance of $\Delta \bar{v}(\tilde{\mathcal{I}}, \mathcal{I}_i)$.

