# OpenReview forum: "Interpreting Global Perturbation Robustness of Image Models using Axiomatic Spectral Importance Decomposition"
_TMLR — Accepted by TMLR_

### Review · Reviewer_PKrD · 2024-05-27

**Summary Of Contributions:**

This paper considered Interpreting Global Perturbation Robustness through the understanding of spectral importance. Specifically, this paper considered the Shapley value in the context of image spectral space to analyze the importance of high-/low- frequency signals.

**Audience:**

Yes

**Claims And Evidence:**

Yes

**Requested Changes:**

- Page 1

I would think the title/abstract should highlight the context of Image modality. This paper proposed a compelling concept for image but this does not imply it’s insightful for all the kinds of data modalities such as tabular, graph and NLP. The current title is too general and gives a wrong impression that it works for all kinds of data modality. Indeed, the whole paper only focuses on image (I believe this is great! It’s just to adjust the paper scope).

- Page 1, Sec 1, Introduction

I have similar concerns with the introduction section. It clearly stated that the main scope is image modelling and robustness. Therefore it should clearly revise the abstract and title.

In paragraph 3, the SNR is introduced in an unnatural manner, where I think it’s better to put it into the background section rather than intro. Besides, Appendix B3 here still seems quite awkward for new audiences.

In the introduction section, I have a feeling that this is written for the reviewers who already had reviewed your paper and checked for the second round. However, for a new reviewer like me, I still feel a bit hard to digest this section. I would recommend a smoother transition for the new audiences.

- Page 5 Sec 3.1

I could not understand why we need to normalize SID?

- Page 6 Sec 3.3

I feel like the information theoretical interpretation could not be counted as a novel contribution. In general, such a technique has been demonstrated in the literature such as [1]. A clarification is required.

[1] Understanding Global Feature ContributionsWith Additive Importance Measures. NeurIPS 2020.

- Page 8

Could you explain a bit more about the role of \beta?

- Page 14

I really like the discussion on the limitation sections. Is it possible to elaborate a bit more on low- high- frequency adversarial attack?

**Strengths And Weaknesses:**

I feel like this paper proposed a very interesting and principled approach to understand the robustness in the context of images. I really like the idea of incorporating the inductive bias on the image itself. Overall, I have no major concerns but some minor points to be addressed. **Please see the request changes for details**

---

### Review · Reviewer_wwaX · 2024-06-09

**Summary Of Contributions:**

The authors present an approach for robustness quantification for computer vision models. Building on work from information theory, eXplainable Artificial Intelligence (XAI), information theory and game theory, the authors derive a model agnostic workflow for perturbing images and evaluating robustness w.r.t. to those perturbations in frequency space. Compared to other XAI approaches that investigate models in terms of single pixels, patches, concepts or entire images, adding the classical spatial frequency decomposition as semantic concepts for XAI is a great idea. And as many perturbations used in nowadays training of computer vision models as well as for XAI purposes, a better understanding of these perturbations w.r.t. their impact on predictive performance is timely and relevant.

While the structure of the manuscript is a bit unconventional (for instance separate subsections of the experiments have mirrored the classical manuscript structure with methods/results/discussion), this actually works well in parts, in other parts I sometimes found it difficult to understand whether this work is about XAI, about robustness or about any of the (very relevant and well tested) empirical questions in the experimental sections. It feels a bit like there is a lot of work that went into this with a theoretical motivation and then the authors looked for an empirical use case. As in: we'd like to do something with information theory and spectral image decompositions, but computer vision models nowadays all work in decompositions induced by neural networks, so we need to somehow draw this connection and find an application of those pre-ImageNetMoment methods)to state-of-the-art applications.
One way of reducing potential confusion would be to highlight directly in the abstract some of these (i think) great empirical findings on the relative robustness of some models w.r.t. some types of perturbations.

**Audience:**

Yes

**Broader Impact Concerns:**

No concerns, on the contrary: Testing for robustness and explaining potential shortcomings is key to reponsible usage of ML models; I'm convinced this work can make a valuable contribution towards better assessment of model robustness.

**Claims And Evidence:**

Yes

**Requested Changes:**

For a better assessment of the pros and cons of the proposed method, it would be great if the authors could:

* one could put a legend in the figures? in fig 9 it seems that the colors are not used consistently, vit is pink in the upper two panels and green in the lower two? ideally one would use the same colormap across all figures, e.g. also in fig. 10, to enable readers to quickly understand which model is which. Also, the sizes of the circles in fig 8/9 seems to be relevant but it's not explained in the figure.
* generally i'd recommend to try and make the figures/captions as self-explanatory as possible; that will help to reduce cognitive friction when processing these central elements of the publication. More concretely, one could for instance guide the attention to the aspects that the authors deem relevant for the reader
* maybe one could elaborate on assumption 3.7 as to where that assumption comes from. It reads a bit like this is an empirical finding that gets explained post-hoc, while it seems more intuitive that this is not a property of the model or the perturbations, but rather due to the empirical distribution of natural images in spatial frequency space.
* in fig  10b - what's shown there on the x and y axis respectively?
* also fig. 10b: it's great that the authors put the research question in that caption, but ideally the caption would contain the answer (or rather: the figure should contain the answer and the caption draws the attention of the reader to the relevant aspects in the figure). I know it's stated in the text clearly and in bold face, but redundancy always helps and having the relevant information at the relevant positions helps, too.
* the manuscript could be strengthened by being more specific / detailed at times. For instance when the authors write: *"efficientnet, can achieve comparable robustness performance"* it would be helpful to mention how robustness was measured in that case.

**Strengths And Weaknesses:**

## Strengths
* Developing solutions for assessing robustness of computer vision models is a timely and highly relevant topic
* Comparing robustness of models empirically is a valuable contribution
* The authors do a great job at embedding their work in the existing literature
* I think it's great that the authors make an effort to explain their results. For instance the aspect that efficient net seems to be special and how that is related to the frequency distribution
* in a way the study helps to connect the pre-ImageNetMoment research to post-ImageNetMoment research. characterizing robustness in terms of spatial frequency decompositions is interesting and helpful
* I'm convinced that the proposed approach is a valuable contribution to a better understanding of robustness of computer vision models;

## Weaknesses
* Some of the figures could be improved w.r.t. detail and content
* I'm wondering whether the interpretability spin is needed / helpful for the narrative - i was at times confused about the goal of the work
* I'm not sure i understand how label noise is related to spectral perturbations of images. I mean, certaintly, both affect robustness, but i wouldn't expect the image to change. I think the finding that *"Models tend to use more non-robust features in the presence of higher label noise within training set."* is very interesting, but as the perturbation on labels is independent of the spectral perturbations it would be helpful to comment on that relationship

---

> ### Author Response · Authors · 2024-06-21
> **We have revised the manuscript as per your requested changes. Sincerely thank you for the constructive comments.**
>
> To reviewer wwaX,
>
> We sincerely thank you for your efforts and the concerns again.
>
> We have revised the manuscript as per your requests. The revisions are highlighted with blue colour.
>
> # Regarding XAI versus robustness.
> > in other parts I sometimes found it difficult to understand whether this work is about XAI, about robustness
>
> We appreciate your concerns. Our work is not a usual XAI paper. The research subject of this paper is on **robustness**. IASIDE **interprets** the **robustness mechanisms** of image models. This emphasis was also advised by previous AE and reviewers.
>
>
> # Regarding "some models w.r.t. some types of perturbations."
> > ...or about any of the (very relevant and well tested) empirical questions in the experimental sections.
> One way of reducing potential confusion would be to highlight directly in the abstract some of these (i think) great empirical findings on the relative robustness of some models w.r.t. some types of perturbations.
>
> IASIDE is a more fundamental method, which can be applied to study the robustness problem in relevant scenarios, for example:
> - answering the concerns in **architectural robustness**,
> - studying the learning dynamics with the presence of **noisy-labels**.
>
> We have demonstrated the above in two case studies.
>
> Also, we have stated the scope in the **Abstract** :
> > ... In this research, we present a model-agnostic interpretability method **to interpret the perturbation robustness mechanisms of image models** ...".
>
> Please refer to the **Abstract** and our **contribution claims.**
>
> # Regarding:
> > It feels a bit like there is a lot of work that went into this with a theoretical motivation and then the authors looked for an empirical use case. As in: we'd like to do something with information theory and spectral image decompositions, but computer vision models nowadays all work in decompositions induced by neural networks, so we need to somehow draw this connection and find an application of those pre-ImageNetMoment methods)to state-of-the-art applications.
>
> ## **Motivations**.
> We appreciate your concerns. This work is motivated by empirical motivations, particularly in that:
> > The spectral signal-to-noise ratios (SNRs) of perturbed images decay with a power-law-like distribution with respect to the frequency.
>
>
>
>
> # Regarding the consistent colormaps of figures.
>
> We agree with you that the colours should be consistent in Figure 8 and 9. We have revised the figures as per your requests.
>
> # Regarding self-explanatory captions.
>
> We agree with your concerns. We have revised the captions as per your request for being more self-explanatory.
>
> In fact, of our earlier revisions,  our captions were in a very **self-explanatory style** but were suggested to make it shorter. To be honest, it is difficult to make the trade-off between **succinct captions** and **self-explanatory captions**. We made efforts to achieve a better trade-off between **succinct captions** and **self-explanatory captions**.
>
>
>
> # Regarding the Assumption 3.7.
>
> We appreciate your concerns. The **Assumption 3.7** stems from our experiments, as shown in **Figure 6**. We have, empirically, observed that: **"Un-trained models do not have spectral preferences"**.
>
> To make self-explanatory, we added text in the caption of **Figure 6** to state that the experiments will be summarised into Assumption 3.7.
>
> # Regarding Figure 10b.
>
> We appreciate your concerns. We have added the axis labels. We also added annotations into the Figure 10b to aid better the readers regarding the research question. We have revised the caption to make it more self-explanatory. We appreciate your help and great suggestion.
>
>
> # Regarding *efficientnet*
>
> > the manuscript could be strengthened by being more specific / detailed at times. For instance when the authors write: "efficientnet, can achieve comparable robustness performance" it would be helpful to mention how robustness was measured in that case.
>
> We agree with you. We have added the the line "*... by error rates on benchmark datasets ...*".
>
> # **Endeavours to understand *EfficientNet***.
> When we examined the results measured by our method **IASIDE**, we noted that **efficientnet** is surprisingly robust, as shown in **Figure 10b** and **Figure 8 + Figure 9**.
>
> Intuitively, **ConvNets** are less robust according to previous literature. We first confirmed that our measuring method has no problem. We want to understand the **cause** behind this phenomenon/outlier. We conducted literature research. We realised the robustness links to the implicit frequency regularization imposed by the **NAS** (Neural Architecture Searching).
>
> We have briefly summarised this investigation results and cited literature in the very short discussion of **Section 4.2 Studying architectural robustness**, which is a case study experiment.

---

> ### Author Response · Authors · 2024-06-21
> **Our concerns and a trade-off between **emphasis** and presenting **more**.**
>
> # Trade-off
> Although we have found many interesting results by using **IASIDE**,  due to the length and emphasis, the previous AE has advised us to narrow the focus.

---

### Review · Reviewer_rJbZ · 2024-06-12

**Summary Of Contributions:**

The authors apply Shapley value theory to quantify the predictive power of different image features. A decomposition of the image into spectral bands provides a set of *players* which are then used to form *spectral coalitions*. Each of those resulting coalitions can be used as input to a (frozen) classifier, by inverting the decomposing transformation to obtain a filtered image without the excluded spectral bands. The value assigned to a coalition is then defined as the gains in terms of the negative cross-entropy obtained when the spectral bands are included compared to when they are excluded. This information gain is established more directly by expressing the mutual information as a sum of the Shannon entropy and the value function, plus additional terms including a KL-divergence between the predicted and ground-truth labels. The proposed value decomposition can be used to derive a new spectral robustness score, which the authors proceed to apply in a series of experiments establishing its correlation with previously proposed robustness metrics, as well as its application to two case studies in robustness evaluation.

**Audience:**

Yes

**Claims And Evidence:**

Yes

**Requested Changes:**

Abstract:
=======
- "model perturbation robustness mechanisms" is a very complex string. Even the simpler "perturbation robustness" still needs to be defined, in contrast to the more familiar notion of adversarial robustness. [R1] Please clarify the scope of the paper by defining "perturbation robustness" first, before discussing the issue of understanding its mechanisms. In doing so, please use simpler sentence structures, e.g., the mechanisms of perturbation robustness of machine learning models.
- Given the confusion about the scope, and the complexity of the language in the beginning of the abstract, I find it extra distracting to prematurely switch attention to the specifics of the approach, let alone its name or acronyms. [R2] Please defer naming the method till after the scope/motivation and a summary of the technical ideas have been explained clearly.

Section 1:
=======
- The authors set out to study robustness "mechanisms" as the first thrust motivating this paper (i.e. existing robustness scores failing to provide further insights into the robustness mechanisms). It would help to [R3] please elaborate on what is meant by robustness mechanisms, perhaps by enumerating different considerations over different stages of the model development cycle. It should be fine to focus on those aspects actually studied in the paper, where a reference to recent surveys or concurrent work may suffice for other such considerations. This would help make it clear how much the progress is achieved along this direction through the presented contributions -- which is not very clear to me right now.
  - I like the study of label-noise impact, presented in Section 4.3, as a more interesting example of a "mechanism" since it touches upon the varied scenarios encountered in the training stage. So, I wonder what other kinds of studies can be designed to probe more aspects of robustness mechanisms.

Section 3
=======
- Although the connection to information theory was advertised as early as the abstract, I see it only appears in the last part of Section 3.3. It may be fine to defer the full proof to the appendix, but [R4] please include at least a high level narrative or a walkthrough a simple numerical example to help make the connection to information theory more clear and accessible.
  - It would be nice to [R5] distinguish the game theory framework (coalitions and Shapley value) from the information theory connection, that arises due to the specific characteristic function used.
  - [R6] please make sure to cite and discuss more of the works on interpretability that utilized Shapley values, rather than just SAGE which is mentioned very briefly. For example:
    - Kumar, Indra, Carlos Scheidegger, Suresh Venkatasubramanian, and Sorelle Friedler. "Shapley Residuals: Quantifying the limits of the Shapley value for explanations." Advances in Neural Information Processing Systems 34 (2021): 26598-26608.
    - Some concurrent work also seems relevant:
      - Herbinger, Julia, Bernd Bischl, and Giuseppe Casalicchio. "Decomposing global feature effects based on feature interactions." arXiv preprint arXiv:2306.00541 (2023).
      - Huang, Xuanxiang, and Joao Marques-Silva. "The inadequacy of Shapley values for explainability." arXiv preprint arXiv:2302.08160 (2023).
- Spectral importance distribution (SID): It would also help to [R7] please clearly define $\Psi$ as a vector in $\mathbb{R}^M$. I would also recommend to reserve the asterisk to optimal values, rather than normalized values which typically use a bar.
- Spectral coalition filtering: I would recommend to [R8] please refer to $\mathbb{T}$ consistently as a mask map or an indicator function - I would personally prefer to call it a mask map.
- It would help to [R9] please clarify that spectral coalition images are "fed into" the classifier models (either trained or un-trained) for inference only, and that the proposed technique does not require training a dedicated model for each coalition.

Section 4:
=======
- The experiment with label noise is quite intriguing. In regards to its conclusion, i.e., the last sentence in bold at the end of Section 4.3, I'm not sure if we're looking at causation or correlation. I wonder if forcing the model to give more weight to the robust features, i.e., lower frequency bands, or completely suppress non-robust features, i.e., high frequency bands, would yield a more robust model. [R9] It would be nice to include such an experiment.
  - More generally, following the previous point, it would be nice to consider using the proposed SID as a training objective. That is, if we have a clear idea of what the SID for a robust model looks like, we can encourage the models we train to have a robust SID profile.

Nitpicking
========
- deploy -> apply

**Strengths And Weaknesses:**

Strengths:
- Overall, the paper is well presented and easy to read.
- Offers a formal approach to (robustness) attributions for each spectral band.
- Sheds new light on known phenomena such as the relative robustness of high/low frequency features. (Fig. 6)
- Sheds new light on relevant questions such as the relative robustness of ViT vs ConvNets and small/large models. (Fig. 10)

Weakness:
- There seems to be a bit of a gap in the presented study, which also leads to a bit of disconnect while reading through the paper, since the presentation (title + intro) focus more heavily on the intended application rather than the new technique and formulations presented. It would have been nice to spend more time on the new techniques and showcase its applications to a more wide range of problems, rather than just perturbation robustness. Indeed, the abstract makes a better case for the techniques developed, compared to the (intro + title), by explaining how they offer a formal tool to quantify the attributions of different spectral bands, which has a wider impact beyond robustness considerations.

---

> ### Author Response · Authors · 2024-06-12
> **We would like to sincerely thank you for your efforts, and the constructive suggestions.**
>
> To Reviewer rJbZ,
>
> We would like to sincerely thank you for your efforts, and the constructive suggestions. We appreciate your great help.
>
> We are now revising the manuscript to address your concerns and requested changes. We will online a revised version and post a rebuttal shortly.
>
> Best and thanks,
>
> Authors of manuscript 2588

---

> ### Comment · Reviewer_rJbZ · 2024-06-13
> **One more remark**
>
> It occurred to me that the decision to only study perturbations up to 10% of the energy of the clean image could be playing a big role in key conclusions, such as the relative robustness of different features. Intuitively, features with low ESD can be more easily shifted, distorted, or overwritten by perturbations of similarly low ESD. The same cannot be said for features with high ESD.
>
> I understand it's typical to consider perturbations of small magnitudes to avoid noticeably changing the image, as in the case of adversarial attacks. Nonetheless, I think it's important to highlight this in the context of the present work.
>
> It would also help to apply the proposed tools to analyze other types of perturbations (adversarial attacks) that don't simply add noise, e.g., unrestricted attacks:
> - Chen, Zhaoyu, Bo Li, Shuang Wu, Kaixun Jiang, Shouhong Ding, and Wenqiang Zhang. "Content-based unrestricted adversarial attack." Advances in Neural Information Processing Systems 36 (2023).
> - Yuan, Shengming, Qilong Zhang, Lianli Gao, Yaya Cheng, and Jingkuan Song. "Natural color fool: Towards boosting black-box unrestricted attacks." Advances in Neural Information Processing Systems 35 (2022): 7546-7560.
> - Also, e.g., Wang, Xiaosen, Zeliang Zhang, and Jianping Zhang. "Structure invariant transformation for better adversarial transferability." In Proceedings of the IEEE/CVF International Conference on Computer Vision, pp. 4607-4619. 2023.

---

> > ### Author Response · Authors · 2024-06-13
> > **Sincerely thank you for these extra constructive suggestions and references.**
> >
> > To Reviewer rJbZ,
> >
> > We would like to sincerely thank you again for the extra constructive suggestions and references.
> >
> > We already started the revisions, rebuttal preparation and the interesting **experiment** you mentioned (low-freq versus high-free). We will present to you in the coming rebuttals and support materials shortly.
> > > We trained models with only high-freq/low-freq signals. The experiments WELL support your feelings in the comments regarding low-/high- freq signals. And also the results further substantiate our assumptions regarding signal robustness.
> >
> > We are now endeavouring to address your concerns and requested changes. This may take some days, as we need to discuss and achieve consensus among all authors.
> >
> > We will online a revised version and post a rebuttal shortly.
> >
> > Best and thanks,
> >
> > Authors of manuscript 2588

---

> ### Author Response · Authors · 2024-06-21
> **Revisions and rebuttals part 1.**
>
> To Reviewer rJbZ,
>
> We sincerely thank you for the efforts and concerns.
>
> We have revised the manuscript as per your requests. The revisions are highlighted with blue colour.
>
> # Our considerations and trade-off in scope.
>
> We have found more interesting results using IASIDE **beyond the presented findings** in this paper. However, we were also advised by AE and reviewers to focus on our emphasis in one paper.
>
> We have been pondering a **trade-off** between ***narrowing** the scope* and ***widening** for presenting more interesting findings*.
>
> Considering that the current manuscript reaches already 16 pages, we would like to present more findings in one or two continuing papers further discussing robustness topic (eg on TMLR).
>
> # Regarding Abstract
>
> ### R1: Scope and definition.
> >... [R1] Please clarify the scope of the paper by defining "**perturbation robustness**" first, before discussing the issue of understanding its mechanisms ...
>
> We agree with your concerns. We have revised the **Abstract** as per your request **R1**. We also have made trade-off to avoid too many details in the**Abstract**, which may distract/confuse our readers.
>
> ### R2: Deferring naming the method.
> > -   Given the confusion about the scope, and the complexity of the language in the beginning of the abstract, I find it extra distracting to prematurely switch attention to the specifics of the approach, let alone its name or acronyms. [R2] Please defer naming the method till after the scope/motivation and a summary of the technical ideas have been explained clearly.
>
> We agree with your concerns by postponing the naming upon motivations are explained clearly. We have revised the **Abstract** by postponing the naming.
>
>
> # Concerns in Section 1
>
> ### R3: Defining robustness mechanisms.
>
> We agree with your concerns regarding the mechanistic interpretations of robustness. We have revised the introduction by adding a paragraph in that:
> - (1) introducing the stages, which may cause robustness problems;
> - (2) clarifying mechanistic interpretations regarding robustness, and its current advances.
>
> Unfortunately, **the global interpretability regarding robustness is a missing piece of the interpretability jigsaw puzzle**. This is because current interpretability methods are not designed to interpret robustness. We have stated this in the introduction by properly citing state-of-the-art works.
>
> ### More scenarios other than the noisy-label showcase.
>
> We appreciate the high assessment regarding the value of our method. Beyond the noisy-label showcase, other experiments using IASIDE are also intriguing, we were advised by previous AE and reviewers **to narrow the scope and put an emphasis** on presentation. We would like to present the remaining results in a different manuscript.

---

> ### Author Response · Authors · 2024-06-21
> **Revisions and rebuttals part 2.**
>
> # Concerns in Section 3
>
> ### R4: Ink in information theory.
> ### R5: Decoupling Shapley value theory and information theory.
>
> > ... [R4] please include at least a high level narrative or a walkthrough a simple numerical example to help make the connection to information theory more clear and accessible.
>
> > ... [R5] distinguish the game theory framework (coalitions and Shapley value) from the information theory connection, that arises due to the specific characteristic function used.
>
> We sincerely appreciate these constructive suggestions in **R4** and **R5**. We agree with you in that: **The connection to information theory is a property by specially designing the characteristic maps** of Shapley value theory.
>
> We have revised regarding the **connection** in the **Introduction: contributions** and **Method: high-level overview**:
> - Putting extra ink to the high-level narrative of information theory etc.
> - Decoupling **Shapley value theory** and **information theory**.
>
>
>
> ### R6: Citing more works utilizing Shapley value theory.
> > ... [R6] please make sure to cite and discuss more of the works on interpretability that utilized Shapley values, rather than just SAGE which is mentioned very briefly...
>
> We appreciate the concerns and the provided references. We have revised the manuscript in the **Section: Related work** by adding more works.
>
> However, we have some concerns:
> 1. Most interpretability methods, applying Shapley value theory framework, are not designed for **global interpretability**. This manuscript is in the scope of **global interpretability**. We would like to include the literature only falling in this scope.
> 2.  Although some methods, eg SAGE, are designed for global interpretability, these methods are not designed to interpret **robustness**.
>
>
> ### R7.1: Clearly defining  $\Psi$ as a vector in $\mathbb{R}^M$,
> ### R7.2: Changing notation ${\cdot}^*$ to $\bar{\cdot}$ for respecting the convention regarding the asterisk .
>
> > Spectral importance distribution (SID): It would also help to [R7] please clearly define $\Psi$ as a vector in $\mathbb{R}^M$. I would also recommend to reserve the asterisk to optimal values, rather than normalized values which typically use a bar.
>
> We agree with you. We have revised the manuscript as per your concerns by:
> - Clearly defining $\Psi$ is a vector in $\mathbb{R}^M$.
> - Changing notation notation ${\cdot}^*$ to $\bar{\cdot}$ for respecting the convention regarding asterisk. The changes relate to the entire manuscript including text and proofs.
>
> ### R8: Referring to $\mathbb{T}$ as "mask map", of the coalitional filtering.
>
> > Spectral coalition filtering: I would recommend to [R8] please refer to $\mathbb{T}$ consistently as a mask map or an indicator function - I would personally prefer to call it a mask map.
>
> We agree with your concerns regarding consistency and readability. We have revised the manuscript by:
> - Referring to $\mathbb{T}$ as "mask map".
>
> ### R9: clarify that spectral coalition images are "fed into" the classifier models
> > It would help to [R9] please clarify that spectral coalition images are "fed into" the classifier models (either trained or un-trained) for inference only, and that the proposed technique does not require training a dedicated model for each coalition.
>
> We agree with your concerns. We have revised the manuscript as per the request. We also added "**Frozen**" in the Figure 4 (framework overview) and revised Figure 5 (coalitional filtering).

---

> ### Author Response · Authors · 2024-06-21
> **Revisions and rebuttals part 3, including the replies to the one more remark.**
>
> # Concerns in Section 4.
>
> ### Extra experiment regarding low-freq/high-freq.
>
> > The experiment with label noise is quite intriguing. In regards to its conclusion, i.e., the last sentence in bold at the end of Section 4.3, I'm not sure if we're looking at causation or correlation. I wonder if forcing the model to give more weight to the robust features, i.e., lower frequency bands, or completely suppress non-robust features, i.e., high frequency bands, would yield a more robust model. [R9] It would be nice to include such an experiment.
>
> We appreciate your concerns and thank you for your nice suggestions. We agree with your opinions in that including an experiment as suggested would further benefit the soundness.
>
> We had conducted the experiments, and presented as the **Figure 4: Role of frequency**.  There are two findings:
> - (1) both high-frequency and low-frequency signals contain sufficient discriminative information to achieve considerable training accuracy;
> - (2) however, models trained with low-frequency signals give models higher robustness compared to high-frequency signals.
>
> Please refer to:
> - Figure 4,
> - and, the first paragraph, in the page 3 of **Introduction**.
>
> ### SID as regularizer.
>
> > More generally, following the previous point, it would be nice to consider using the proposed SID as a training objective. That is, if we have a clear idea of what the SID for a robust model looks like, we can encourage the models we train to have a robust SID profile.
>
> We had conducted the experiments earlier. Due to the length and focus, we would like to present in next paper following this manuscript including more interesting results and analyses.
>
> Your feeling is correct, penalising with SID can further improve the  robustness of models.
>
> However, the **cost** is on longer training steps, as penalising implies that some information will be suppressed. Models need more training steps to converge.
>
> Also, we need to be very careful to trade-off what are "high-" and "low-" frequency signals. It is a relative problem, we may need smarter strategy to integrate the SID-regularizer into existing optimization frameworks.
>
>
> # Regarding motivation experiment.
>
> We appreciate your concerns regarding our motivation experiment (Figure 2).
>
> > It occurred to me that the decision to only study perturbations up to 10% of the energy of the clean image could be playing a big role in key conclusions, such as the relative robustness of different features. Intuitively, features with low ESD can be more easily shifted, distorted, or overwritten by perturbations of similarly low ESD. The same cannot be said for features with high ESD.
>
> We agree with you. Yes, the intuition acts as the second motivation, ie the SNR as shown in Figure 2.
>
> We actually measured the SNR with various energy levels. The distributions are the same. The only difference is the position of the **0 dB** (on x-axis). When we increase the perturbation levels, the 0DB moves towards right side of the x-axis.
>
>
>
> > I understand it's typical to consider perturbations of small magnitudes to avoid noticeably changing the image, as in the case of adversarial attacks. Nonetheless, I think it's important to highlight this in the context of the present work.
>
> We appreciate your concerns. We have stated in the manuscript as:
> > We set the energy of perturbations to $10\%$ of the energy of the clean image for a fair comparison.
>
> In fact, **10%** is a huge energy magnitude. For example, when the perturbation with energy magnitude **10%** presents, the top-1 accuracy of a resnet50 will drop from 76.15% to less than 11%, pre-trained on ImageNet.
>
> The SNR distributions remain the same with various energy levels. The only difference is the 0-DB position will shift.
>
> Please also refer to the new results in Figure 3.
>
>
> > It would also help to apply the proposed tools to analyze other types of perturbations (adversarial attacks) that don't simply add noise, e.g., unrestricted attacks:
>
> We have included adversarial attacks, including FGSM and PGD, in the motivation experiments of Figure 2.
>
> The perturbations from FGSM and PGD are not obtained by adding noise. The perturbations are obtained from a resnet50.

---

> ### Comment · Reviewer_rJbZ · 2024-06-22
> **Final comments on June 21st version**
>
> Presentation:
> ==========
> - Abstract:
>   - *extensive experiments over a variety of vision foundation models*: I can see the word foundation only appears twice, this one and the title of the reference to [Awais et al, 2023]. Please [FR1] revise this high level account of the experiments presented to accurately reflect the actual models used in the present submission. I, for one, don't consider any transformer-based model to be a foundation model, depending on how the model in question was trained. So, please [FR2] only use the keyword "foundation model" as widely understood by the community.
>
> Nitpicking:
> ========
> - Section 1:
>   - *Robustness mechanisms are the answer to the question*: recommend to [FR3] please rephrase this as "The study of robustness mechanisms (aim to answer / are motivated by) the question"
> - Section 4:
>   - *The answers for the two questions by using I-ASIDE are yes*: recommend to [FR4] please rephrase this as " the answers/conclusions suggested by using I-ASIDE ..."
> - Section 5:
>   - *Global input feature importance analysis, often by using Shapley value theory framework, answers the question*: recommend to [FR5] please rephrase this as "aims/attempts to answer the question"
>
> It's only fair that I justify those nitpicking requests. In my view, there are many ambiguities even in the statement of the questions, and more importantly in the underlying mechanisms of representation power and different training strategies. It is true that there are common assumptions and practices within the community that favor particular stances to those ambiguous points, but it is more accurate, and hopefully beneficial, to recognize those ambiguities for what they are and leave room for some healthy skepticism.

---

> > ### Author Response · Authors · 2024-06-24
> > **We sincerely thank you for the great efforts and constructive suggestions.**
> >
> > To Reviewer rJbZ,
> >
> >
> > We have seen great efforts you have on our manuscript for making the manuscript perfect.
> > We sincerely thank you again for your great efforts and the constructive suggestions.
> >
> > Our authors fully agree with your views regarding the **nitpicking requests**. We truly appreciate comments like these from reviewers. To be honest, we also believe these nice comments **definitely** further improve the quality of the presentation. The enhancement(s) would benefit the entire **community** in a heathy way. As authors, we also enjoy in engaging with the research community in this way. Thank you again.
> > > It's only fair that I justify those nitpicking requests. In my view, there are many ambiguities even in the statement of the questions, and more importantly in the underlying mechanisms of representation power and different training strategies. It is true that there are common assumptions and practices within the community that favor particular stances to those ambiguous points, but it is more accurate, and hopefully beneficial, to recognize those ambiguities for what they are and leave room for some healthy skepticism.
> >
> > We have revised the manuscript as per your requests. The revisions are highlighted with blue colour.
> >
> > # Regarding FR1.
> > > Please [FR1] revise this high level account of the experiments presented to accurately reflect the actual models used in the present submission.
> >
> > We appreciate your concerns and agree with you. We have revised the **Abstract** by listing the models we have used in this manuscript.
> >
> > # Regarding FR2.
> >
> > >  I, for one, don't consider any transformer- based model to be a foundation model, depending on how the model in question was trained.  So, please [FR2] only use the keyword "foundation model" as widely understood by the community.
> >
> > We appreciate your concerns and agree with you. We have revised the **Abstract** and **Introduction** as per the request **FR2**. We also corrected the citation regarding this.
> >
> >
> > # Regarding FR3.
> > >   Robustness mechanisms are the answer to the question: recommend to [FR3] please rephrase this as "The study of robustness mechanisms (aim to answer / are motivated by) the question"
> >
> > We appreciate your concerns and agree with you. We rephrased the sentences by using your suggestion. We believe the rephrasing could further improve the clearness of the definition. We thank you again.
> >
> > # Regarding FR4.
> >
> > > The answers for the two questions by using I-ASIDE are yes: recommend to [FR4] please rephrase this as " the answers/conclusions suggested by using I-ASIDE ..."
> >
> > We appreciate your concerns and agree with you. We rephrased the sentence as per the request.
> >
> > # Regarding FR5.
> >
> > > Global input feature importance analysis, often by using Shapley value theory framework, answers the question: recommend to [FR5] please rephrase this as "aims/attempts to answer the question"
> >
> > We appreciate your concerns and agree with you. We rephrased the sentence as per the request.
> >
> > # For the final version.
> > We would like to thank you again for your great efforts. We will further polish the manuscript in the final version.
> >
> > Best.
> >
> > Authors.

---

> ### Author Response · Authors · 2024-06-23
> **We would like to thank you again for the constructive comments. We are now working on the revisions as per your requests.**
>
> To Reviewer rJbZ,
>
> We would like to sincerely thank you again for the efforts and constructive suggestions.
>
> We agree with you in terms of the requested changes in the **Abstract**.
> We also agree with you in terms of the request changes regarding the rephrases.
>
> We are now revising the manuscript to address the final concerns and requested changes.
> Once we reach a consensus among our authors, we will online a revised version and post a rebuttal shortly.
>
> Great thanks again.
>
> Best,
>
> Authors.

---

### Decision · Action_Editor_63bc · 2024-07-16

**Recommendation:** Accept as is

**Comment:**

In this paper, the authors proposed a method I-ASIDE to evaluate the robustness of image models against frequency distribution.
I-ASIDE divides the input image's frequency into bins, masks them, and assesses the contribution of each frequency bin using the Shapley value.

The authors revealed through experiments that image models tend to focus on low-frequency components.
Additionally, they reported that image models with a higher bias towards frequency contribution are more vulnerable to adversarial noise and other such perturbations.
The proposal of I-ASIDE for investigating frequency contributions and the insights into the relationship between these frequency components and the robustness of image models are highly intriguing.

The authors appropriately revised the paper to clarify its message through communication with the reviewers.
As a result, all three reviewers agreed to accept this paper.

**Audience:**

The frequency dependence and robustness of image models are crucial topics in machine learning, especially in the field of image classification. This paper is within the scope of TMLR.

**Claims And Evidence:**

In this study, the authors proposed a method I-ASIDE to evaluate the robustness of image models against frequency distribution.
The authors validate the effectiveness of their proposed method and its implications through comparative experiments on popular image models.
These results seems to be convincing.

---

> ### Author Response · Authors · 2024-07-17
> **Sincerely thank the great help from reviewers and AE 63bc.**
>
> Dear AE 63bc, reviewer rJbZ, reviewer  wwaX, reviewer  PKrD, previous AE and reviewers,
>
> We would like to sincerely thank you for the valuable help and great efforts.
>
> As the authors, we have seen that the quality of the manuscript has been greatly improved as the constructive reviews. We truly enjoyed in the engaging with the research community. We appreciate the constructive valuable help again.
>
> Authors of manuscript 2588